# Quantification of gene expression patterns to reveal the origins of abnormal morphogenesis

Neus Martínez-Abadías[1,2,3]*, Roger Mateu Estivill[4], Jaume Sastre Tomas[5], Susan Motch Perrine[6], Melissa Yoon[6], Alexandre Robert-Moreno[1,2,3], Jim Swoger[1,2,3], Lucia Russo[1,2], Kazuhiko Kawasaki[6], Joan Richtsmeier[6], James Sharpe[1,2,3,7]*

[1]Centre for Genomic Regulation, The Barcelona Institute for Science and Technology, Barcelona, Spain; [2]Universitat Pompeu Fabra, Barcelona, Spain; [3]EMBL Barcelona, European Molecular Biology Laboratory, Barcelona, Spain; [4]Universitat de Barcelona, Barcelona, Spain; [5]Universitat de les Illes Balears (UIB), Palma de Mallorca, Spain; [6]Pennsylvania State University, Pennsylvania, United States; [7]Institució Catalana de Recerca i Estudis Avançats, Barcelona, Spain

**Abstract** The earliest developmental origins of dysmorphologies are poorly understood in many congenital diseases. They often remain elusive because the first signs of genetic misregulation may initiate as subtle changes in gene expression, which are hard to detect and can be obscured later in development by secondary effects. Here, we develop a method to trace back the origins of phenotypic abnormalities by accurately quantifying the 3D spatial distribution of gene expression domains in developing organs. By applying Geometric Morphometrics to 3D gene expression data obtained by Optical Projection Tomography, we determined that our approach is sensitive enough to find regulatory abnormalities that have never been detected previously. We identified subtle but significant differences in the gene expression of a downstream target of a *Fgfr2* mutation associated with Apert syndrome, demonstrating that these mouse models can further our understanding of limb defects in the human condition. Our method can be applied to different organ systems and models to investigate the etiology of malformations.

DOI: https://doi.org/10.7554/eLife.36405.001

**\*For correspondence:**
neus.martinez@embl.es (NM-A);
james.sharpe@embl.es (JS)

**Competing interests:** The authors declare that no competing interests exist.

## Introduction

Morphogenesis is guided by dynamic spatio-temporal regulation of gene expression patterns (*Chan et al., 2017*; *Andrey and Mundlos, 2017*), the zones within tissues where genes are expressed during specific periods in development. Critical changes in the space, time or intensity of gene expression patterns can result in organ malformation, with reduced or even loss of function. These errors of morphogenesis, which occur in approximately 3% of live births (*Toxicology NRC C on D, 2000*), are induced by environmental and/or genetic insults that alter the normal process of development. The common approach to revealing the origin of these alterations is to look for phenotypic abnormalities in animal models and to visually assess the overall patterns of gene expression. This qualitative approach has not, however, been able to identify the earliest signs of dysmorphogenesis in many diseases because the first changes in the gene expression patterns may be subtle, limited in time, and later abnormalities may obscure the original genetic cause. To reveal the primary etiology of congenital malformations, more rigorous methods are needed to quantify the three-dimensional (3D) phenotypes of gene expression patterns. A useful tool for tracing back development should be able to perform a quantitative statistical comparison of normal and disease-altered

**eLife digest** Our development in the womb is complex. Genes need to switch on and off in a precise order, controlling the activity of millions of cells as they work together to form different tissues. For everything to happen smoothly, cells must use instructions provided by each gene exactly at the correct moment and in the correct place. In this biological assembly line, the slightest change can lead to a defect.

Certain genetic mutations can change when and where cells use particular genes, and this can cause errors in development. These kinds of mutations are a common cause of birth defects, but we cannot always pinpoint how they begin. For example, a single mutation in a gene called FGFR2 causes malformations in the head, the heart and the limbs in a rare disease called Apert syndrome.

The first signs that development has gone wrong can be subtle changes in the use of certain genes, impossible to detect with standard methods. As development continues, other processes can mask the impact of problems with certain genes. Ultimately, changes alter the shape of the developing embryo.

Genetically engineered mouse models can mimic the gene defects that cause disease in humans. But current methods are not sensitive enough to detect the very first signs of defects. Now, Martínez-Abadías et al. developed a new method to detect these subtle changes and reveal the precise moment when development starts to go wrong.

In mice, a specific mutation in the FGFR2 gene affects the activity of a series of other genes. To track the levels of one of these genes, Martínez-Abadías et al. marked mouse embryos using a chemical label. Scanning the embryos then revealed the pattern of the cells using the gene during the earliest stages of development. In mice carrying a mutation in the FGFR2 gene, subtle changes in gene expression began just a few hours after their limbs start to develop. But it took another half a day to see the effects of these changes on the shape and size of the growing limbs. This approach revealed changes in gene expression before any problems with development were visible by eye.

Tracking subtle changes in the way cells use genes could allow us to detect the origins of embryo malformations before they appear, pointing at the best moment to start a treatment. With further development, the model could extend to other genes, proteins, animal models and diseases.
DOI: https://doi.org/10.7554/eLife.36405.002

embryogenesis and to detect the earliest signs of genetic misregulation leading to organ malformation.

The shape of developing organs is regulated by gene activity in space and time (*Andrey and Mundlos, 2017*). Cascades of gene regulatory networks provide the detailed instructions necessary to organize cell behavior and to orchestrate tissue growth and differentiation. Gene expression patterns can be readily mapped within tissues in a true 3D framework combining whole-mount-in-situ hybridization (WMISH) (*Rosen and Beddington, 1993*) with Optical Projection tomography (OPT) (*Sharpe et al., 2002*). WMISH is a standard molecular technique for detecting the expression of a specific gene using a labeled complementary RNA probe (*de la Pompa et al., 1997*; *Correia and Conlon, 2001*), and OPT is a mesoscopic imaging procedure that can produce high-resolution 3D reconstructions of whole developing embryos processed by WMISH (*Sharpe, 2003*; *Boot et al., 2008*). These technologies represented breakthroughs in developmental biology and have provided invaluable qualitative insights into gene function and development (*Sharpe, 2003*). However, methods for quantifying the 3D distributions of gene expression in a systematic, objective manner are still lacking.

Expanding the potential of OPT from *qualitative* to *quantitative* analysis of gene expression patterns is challenging. Gene expression is characterized by highly dynamic patterns, with fast rates of change and fuzzy boundaries that usually do not correspond to well-defined anatomical structures but rather to tissue regions where cells have dynamically up- and downregulated genes. Gene expression patterns have rarely been quantified (*Jernvall et al., 2000*; *Airey et al., 2006*; *Salazar-Ciudad and Jernvall, 2010*; *Mayer et al., 2014*; *Hu et al., 2015*; *Xu et al., 2015*; *Martínez-Abadías et al., 2016*). We propose to quantify the shape of developing organs in association with their underlying gene expression patterns by applying Geometric Morphometrics (GM), a set of

statistical tools for measuring and comparing shapes with increased precision and efficiency (*James Rohlf and Marcus, 1993*; *Klingenberg, 2002*; *Klingenberg, 2010*; *Adams et al., 2013*; *Hallgrimsson et al., 2015*). Previous attempts had analyzed the phenotypic and gene expression patterns of variation independently (*Jernvall et al., 2000*), using different morphometric methods (*Hu et al., 2015*; *Xu et al., 2015*) or were restricted to two-dimensional analyses (*Martínez-Abadías et al., 2016*). Therefore, the current study is the first to combine OPT and GM to characterize the shape of gene expression patterns quantitatively in 3D and to associate these changes to phenotypic changes. Thus, this approach provides the ability to replace qualitative observations with the quantification of subtle yet significant biological differences that underlie the processes through which morphogenesis is altered by disease.

Here, we illustrate how our method can reveal the genetic origin of developmental defects by investigating limb malformations in Apert syndrome [OMIM 101200]. Apert syndrome is a rare congenital disease, with a disease prevalence of 15–16 per million live births, that is characterized by cranial, neural, limb, and visceral malformations (*Cohen and MacLean, 2000*). Over 99% of Apert cases are associated with one of two missense mutations, S252W or P253R, in the Fibroblast Growth Factor Receptor 2 (FGFR2) (*Wilkie et al., 1995*; *Park et al., 1995*). The mutations occur on neighboring amino acids on the linker region between the second and third extracellular immunoglobulin domains of FGFR2, and alter the ligand-binding specificity of the receptors (*Yu et al., 2000*; *Yu and Ornitz, 2001*). Thus, the FGF receptors are activated inappropriately, altering the entire FGF/FGFR signaling pathway and causing dysmorphologies of different organs and systems (*McIntosh et al., 2000*). Apert syndrome shares craniofacial dysmorphologies with other craniosynostosis syndromes but is differentiated on the basis of limb defects of the fore- and hindlimb digits. The craniofacial dysmorphology of Apert syndrome (i.e. premature closure of cranial sutures and patent anterior fontanelle associated with atypical head shape, midfacial retrusion and palatal defects) (*Cohen and MacLean, 2000*) has been intensively investigated, especially because mouse models show cranial phenotypes that correspond with the human condition (*Holmes et al., 2009*; *Martínez-Abadías et al., 2010*; *Holmes and Basilico, 2012*; *Hill et al., 2013*; *Heuzé et al., 2014*). The associated limb defects are less well studied, however, in part because mouse models for Apert syndrome present only subtle limb anomalies (*Chen et al., 2003*; *Wang et al., 2005*; *Wang et al., 2010*), even in mice carrying the P253R mutation (*Wang et al., 2010*) which is associated with the more severe limb malformations in Apert syndrome (*Slaney et al., 1996*; *von Gernet et al., 2000*).

Here, we present precise phenotyping of the limbs of newborn and embryonic specimens of the *Fgfr2⁺/P253R* Apert syndrome mouse model, and reveal significant differences that can be traced to as early as one day after the initiation of limb development. To explore the molecular basis of these initial signs of limb dysmorphology, we applied our method combining OPT and GM to assess the expression pattern of a downstream target of Fgf signaling, *Dusp6*. We chose to assess *Dusp6* because it is well-documented as a direct target of Fgf signaling (*Kawakami et al., 2003*; *Li et al., 2007*; *Ekerot et al., 2008*), and because, in addition to being a relevant gene for limb morphogenesis, it is also important for facial, brain and heart development (*Kawakami et al., 2003*; *Li et al., 2007*; *Maillet et al., 2008*). Therefore, *Dusp6* was an ideal candidate for our proof-of-concept study. Our quantitative analyses demonstrate that the Apert syndrome *Fgfr2 P253R* mutation induces changes in the expression pattern of *Dusp6* and that these genetic changes are associated with significant phenotypic alterations. These results provide insight into the origins of limb malformations in Apert syndrome.

## Results

### Apert syndrome mice present limb malformations at birth

Previous studies have reported that most Apert syndrome mice do not show obvious abnormalities of the limbs (*Chen et al., 2003*), and thus focused their molecular analyses on the skull (*Wang et al., 2010*). Histopathological analyses in Apert syndrome mice revealed overall limb shortening resulting from abnormal osteogenic differentiation, but no signs of limb disproportion or syndactyly (*Wang et al., 2005*; *Wang et al., 2010*). Syndactyly has only been reported in three specimens of an outbred knock-in *Fgfr2⁺/P253R* model (*Yin et al., 2008*), but no experimental analyses were

performed to further explain why the FGFR2 mutation did not affect limb development in all the specimens within the sample.

As a further test of whether or not the *Fgfr2 P253R* mutation affects limb development in mice, we first performed an extensive quantitative analysis of the size and shape of individual forelimb bones using data from high resolution microCT images of newborn (P0) mutant and unaffected littermates (*Figure 1A–F*). Our results revealed many more significant differences between P0 unaffected and *Fgfr2^{+/P253R}* mutant littermates than previously reported. We found that the humerus, radius and ulna were statistically significantly shorter in length but had increased bone volumes in *Fgfr2^{+/P253R}* mutant mice in comparison to unaffected littermates (*Table 1* and *Figure 1G*). More localized size differences were detected in the bones derived from the autopod that give rise to the hands. The distal phalanx of digit I, the proximal phalanx of digit V, and metacarpals II, III and IV were significantly longer in *Fgfr2^{+/P253R}* mutant mice (*Table 1* and *Figure 1G*). In contrast, the proximal phalanx of digit III was shorter and lower in bone volume in *Fgfr2^{+/P253R}* Apert syndrome mice relative to unaffected littermates (*Table 1* and *Figure 1G*). The scapula and the clavicle, the bones that form the shoulder girdle, were also significantly affected: the scapula was longer, the clavicle was shorter and both bones showed increased bone volumes in *Fgfr2^{+/P253R}* Apert syndrome mice (*Table 1* and *Figure 1G*) compared to unaffected littermates.

The Principal Components Analysis (PCA) based on the shape of the humerus did not show marked shape differences between unaffected and *Fgfr2^{+/P253R}* Apert syndrome mice (*Figure 1H*). However, the PCA of the scapula indicated a clear morphological differentiation between these two groups (*Figure 1I*). The scapula of *Fgfr2^{+/P253R}* Apert syndrome mice presented a more robust phenotype, with wider and longer scapulae, in comparison to that of their unaffected littermates.

Overall, these size and shape differences demonstrate that *Fgfr2^{+/P253R}* Apert syndrome mice present widespread and significant limb dysmorphologies at P0 that were not previously reported and would not have been revealed without microCT scanning and quantitative statistical testing. Some defects, such as shoulder anomalies and short humeri, have a direct correspondence with the human phenotype (*Park et al., 1995*). In newborn mice, however, we did not detect any clear sign of syndactyly, which is the most prominent limb defect in people with Apert syndrome (*Cohen and Kreiborg, 1995*; *Holten et al., 1997*). As the forelimb of mice is not yet completely ossified at P0 and because *Fgfr2^{+/P253R}* mutant littermates die shortly after birth, we could not assess whether other limb abnormalities might appear later in development.

## A quantitative morphometric method to assess embryonic gene expression patterns

To determine the earliest developmental basis of the limb anomalies quantified in newborn mice, we developed a quantitative method to explore early embryonic limb development. First, to visualize the expression pattern of a downstream target of *Fgfr2,* we obtained OPT scans of *Fgfr2^{+/P253R}* Apert syndrome mouse embryos that had been analyzed with WMISH to reveal *Dusp6* expression (*Figure 2*). Qualitative assessment of the 3D reconstructions showed that *Dusp6* was widely expressed throughout the embryo from embryonic day (E)10.5 to E11.5, with highest intensity in the limbs, the head and the somites (*Figure 2*). *Dusp6* was also expressed in the heart with moderate intensity. By comparing the distribution of the *Dusp6* gene expression pattern visually, it was possible to distinguish between embryos at the E10.5 and the E11.5 stages of development. At E10.5, *Dusp6* was prominently expressed in the facial prominences and in the forming somites along most of the craniocaudal segment, whereas at E11.5, the expression of *Dusp6* was more widespread in the brain and restricted to the caudal somites. Focusing on the limbs, the expression of *Dusp6* at the two different stages was also readily distinguishable, with *Dusp6* expression domains thinning into a more extended domain along the limb outline as the limb buds grow from E10.5 to E11.5 (*Figure 2*). The limb bud expression patterns of *Fgfr2^{+/P253R}* mutant and unaffected littermates were not distinguishable because of the large amount of developmental variation within litters (*Figure 2*).

Quantitative testing was thus required to allow the more accurate evaluation of limb alterations that are potentially associated with Apert syndrome but visually undetectable. We developed a method for 3D shape analysis of the limb and the associated gene expression pattern of *Dusp6* (*Figure 3*). This protocol enabled us to determine differences in limb size and shape between genotype groups and to assess whether these phenotypic differences are associated with altered gene expression patterns (*Figure 3*). Our approach uses GM methods to measure directly the limb anatomy and

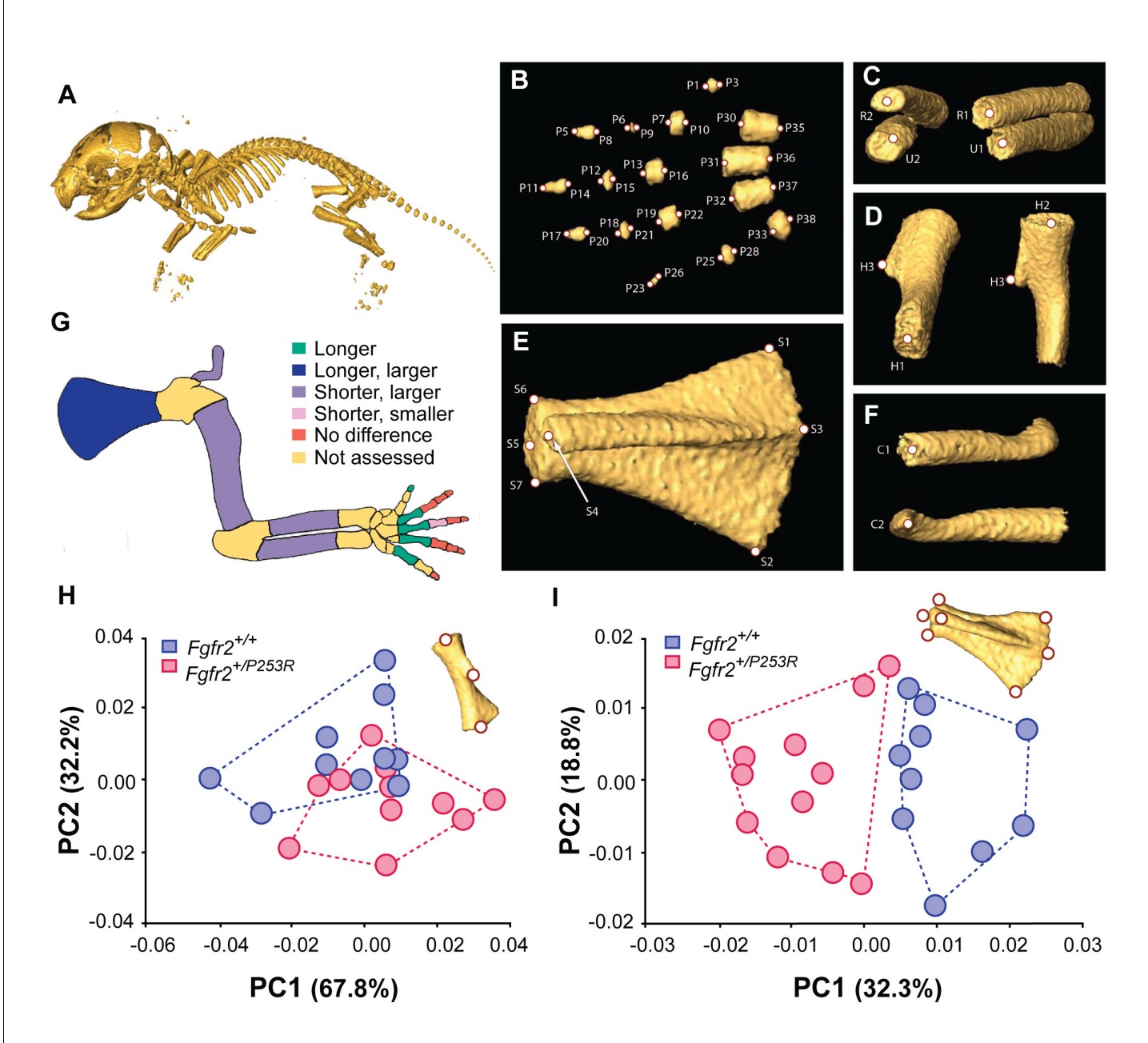

**Figure 1.** Quantitative size and shape comparison of forelimb bones in *Fgfr2^{+/P253R}* newborn mice (P0) and unaffected littermates. (**A**) Mouse skeleton at P0. 3D isosurface reconstruction of the skeleton of an unaffected littermate obtained from a high-resolution μCT scan. (**B–F**) Anatomical landmarks recorded on microCT scans of Apert syndrome mice at P0. (**B**) Autopod (hand). Landmarks were recorded at the midpoint of the proximal and distal tips of the distal, mid and proximal phalanges (P1–P28) and the metacarpals (P29–P38). Proximal phalanx I, middle phalanx V, and metacarpal I are not displayed because these bones have not yet mineralized at P0 and could not be visualized in many specimens. (**C**) Zeugopod. Landmarks were recorded at the midpoint of the proximal and distal tips of the radius (R1–R2) and ulna (U1–U2). (**D**) Stylopod. Landmarks were recorded at the midpoint of the proximal and distal tips of the humerus (H1–H2), as well as at the tip of the deltoid process (H3) (*Figure 1—source data 1*). (**E**) Scapula. Landmarks were recorded at the most superior and inferior lateral points of the scapula (S1–S2), the most posterior point of the spine (S3), the most antero-medial point of the acromion process (S4) and the medial, superior and inferior points of the glenoid cavity (S5–S7) (*Figure 1—source data 2*). (**F**) Clavicle. Landmarks were recorded at the medial point of the sternal and the acromial ends (C1–C2). (**G**) Length and volume differences in the forelimbs of Apert syndrome mouse models. Schematic representation of the forelimb of a P0 mouse showing, in different colors, statistically significant differences in bone length and volume (as measured by two-tailed one-way ANOVA or Mann-Whitney U-test) in *Fgfr2^{+/P253R}* mice and unaffected littermates, as specified in *Table 1*. Longer/shorter refers to length, whereas larger/smaller refers to volume. (**H, I**) Shape differences in the forelimbs of Apert syndrome mouse models. Scatterplots of PC1 and PC2 scores based on Procrustes analysis of anatomical landmark locations representing the

*Figure 1 continued on next page*

*Figure 1 continued*

shape of the left humerus (H) and the left scapula (I) of unaffected (N = 10) and mutant (N = 12) littermates of Apert syndrome mouse models (*Figure 1—source data 1* and *Figure 1—source data 2*). Convex hulls represent the range of variation within each group of mice.

DOI: https://doi.org/10.7554/eLife.36405.003

The following source data is available for figure 1:

**Source data 1.** Source files for humerus shape data.

DOI: https://doi.org/10.7554/eLife.36405.004

**Source data 2.** Source files for scapula shape data.

DOI: https://doi.org/10.7554/eLife.36405.005

gene expression domains segmented from the 3D reconstructions of the embryo OPT scans. As expression of *Dusp6* showed a fuzzy spatial gradient, multiple thresholding was used to define a consistently high gene expression pattern (*Figure 3*, steps from 1 to 5). After manual and semiautomatic recording of the 3D coordinates of landmarks on the surfaces of the limb and the gene expression domains blinded to group allocation (*Figure 3*, step 6; *Video 1*), multivariate statistical analyses were performed to explore shape and size variation and covariation patterns for the limb morphology and the *Dusp6* domain (*Figure 3*, steps 7 and 8).

**Table 1.** Quantitative comparison of mean bone lengths and volumes.

Means were computed as the average lengths or volumes of the right and the left bones for *Fgfr2*$^{+/P253R}$ mutant (N = 12) and *Fgfr2*$^{+/+}$ unaffected (N = 10) mice. Data for proximal phalanx I, middle phalanx V, and metacarpal I are not listed because these bones were not present in all of the specimens, as they may not yet be developed at P0. Statistically significant differences as determined by two-tailed one-way ANOVA or Mann-Whitney U-tests are marked with * (*P-value*<0.05). Statistically significant differences after Bonferroni correction are indicated with **.

| | | Mean bone length (mm ± SD) | | | Mean bone volume (mm³ ± SD) | | |
|---|---|---|---|---|---|---|---|
| | Bone | *Fgfr2*$^{+/+}$ | *Fgfr2*$^{+/P253R}$ | *P-value* | *Fgfr2*$^{+/+}$ | *Fgfr2*$^{+/P253R}$ | *P-value* |
| Autopod | Distal phalanx I | 0.11 ± 0.03 | 0.13 ± 0.02 | 0.004** | 0.0015 ± 0.0008 | 0.0015 ± 0.0008 | 0.86 |
| | Distal phalanx II | 0.22 ± 0.03 | 0.21 ± 0.03 | 0.128 | 0.0021 ± 0.0007 | 0.0022 ± 0.0008 | 0.659 |
| | Middle phalanx II | 0.004 ± 0.003 | 0.01 ± 0.004 | 0.336 | 0.00002 ± 0.00002 | 0.00003 ± 0.00002 | 0.759 |
| | Proximal phalanx II | 0.15 ± 0.01 | 0.14 ± 0.01 | 0.057 | 0.0054 ± 0.0011 | 0.0053 ± 0.0014 | 0.82 |
| | Distal phalanx III | 0.26 ± 0.006 | 0.24 ± 0.01 | 0.068 | 0.0034 ± 0.0008 | 0.0033 ± 0.0017 | 0.796 |
| | Middle phalanx III | 0.10 ± 0.006 | 0.10 ± 0.006 | 0.243 | 0.0019 ± 0.0009 | 0.0022 ± 0.0013 | 0.436 |
| | Proximal phalanx III | 0.204 ± 0.02 | 0.18 ± 0.03 | 0.001** | 0.0087 ± 0.0003 | 0.0073 ± 0.0004 | 0.016* |
| | Distal phalanx IV | 0.24 ± 0.006 | 0.19 ± 0.02 | 0.061 | 0.0024 ± 0.0009 | 0.0024 ± 0.0014 | 0.923 |
| | Middle phalanx IV | 0.06 ± 0.009 | 0.07 ± 0.01 | 0.338 | 0.0009 ± 0.0002 | 0.0014 ± 0.0002 | 0.24 |
| | Proximal phalanx IV | 0.20 ± 0.01 | 0.20 ± 0.02 | 0.412 | 0.0080 ± 0.0016 | 0.0075 ± 0.0021 | 0.358 |
| | Distal phalanx V | 0.09 ± 0.01 | 0.08 ± 0.01 | 0.832 | 0.0005 ± 0.00007 | 0.0006 ± 0.0001 | 0.769 |
| | Proximal phalanx V | 0.14 ± 0.01 | 0.15 ± 0.02 | 0.027* | 0.0031 ± 0.0007 | 0.0036 ± 0.0012 | 0.089 |
| | Metacarpal II | 0.39 ± 0.02 | 0.41 ± 0.02 | 0.034* | 0.0270 ± 0.0028 | 0.0265 ± 0.0034 | 0.586 |
| | Metacarpal III | 0.49 ± 0.03 | 0.52 ± 0.03 | 0.004* | 0.0398 ± 0.0043 | 0.0382 ± 0.0062 | 0.342 |
| | Metacarpal IV | 0.43 ± 0.02 | 0.45 ± 0.03 | 0.011* | 0.0312 ± 0.0033 | 0.0291 ± 0.0048 | 0.114 |
| | Metacarpal V | 0.22 ± 0.01 | 0.22 ± 0.02 | 0.571 | 0.0120 ± 0.0019 | 0.0124 ± 0.0023 | 0.458 |
| Zeugopod | Radius | 2.29 ± 0.02 | 2.22 ± 0.01 | 0.003* | 0.3187 ± 0.0292 | 0.3558 ± 0.0334 | 0.001** |
| | Ulna | 2.76 ± 0.08 | 2.64 ± 0.07 | 0.000** | 0.4999 ± 0.0417 | 0.5366 ± 0.0766 | 0.010* |
| Stylopod | Humerus | 1.65 ± 0.01 | 1.62 ± 0.007 | 0.007* | 0.9950 ± 0.0764 | 1.1444 ± 0.0649 | 0.001** |
| Not derived from limb bud | Scapula | 2.69 ± 0.08 | 2.75 ± 0.06 | 0.013* | 1.0757 ± 0.0733 | 1.2184 ± 0.0777 | 0.001** |
| | Clavicle | 2.47 ± 0.06 | 2.34 ± 0.09 | 0.001** | 0.2214 ± 0.0147 | 0.2811 ± 0.0232 | 0.001** |

DOI: https://doi.org/10.7554/eLife.36405.006

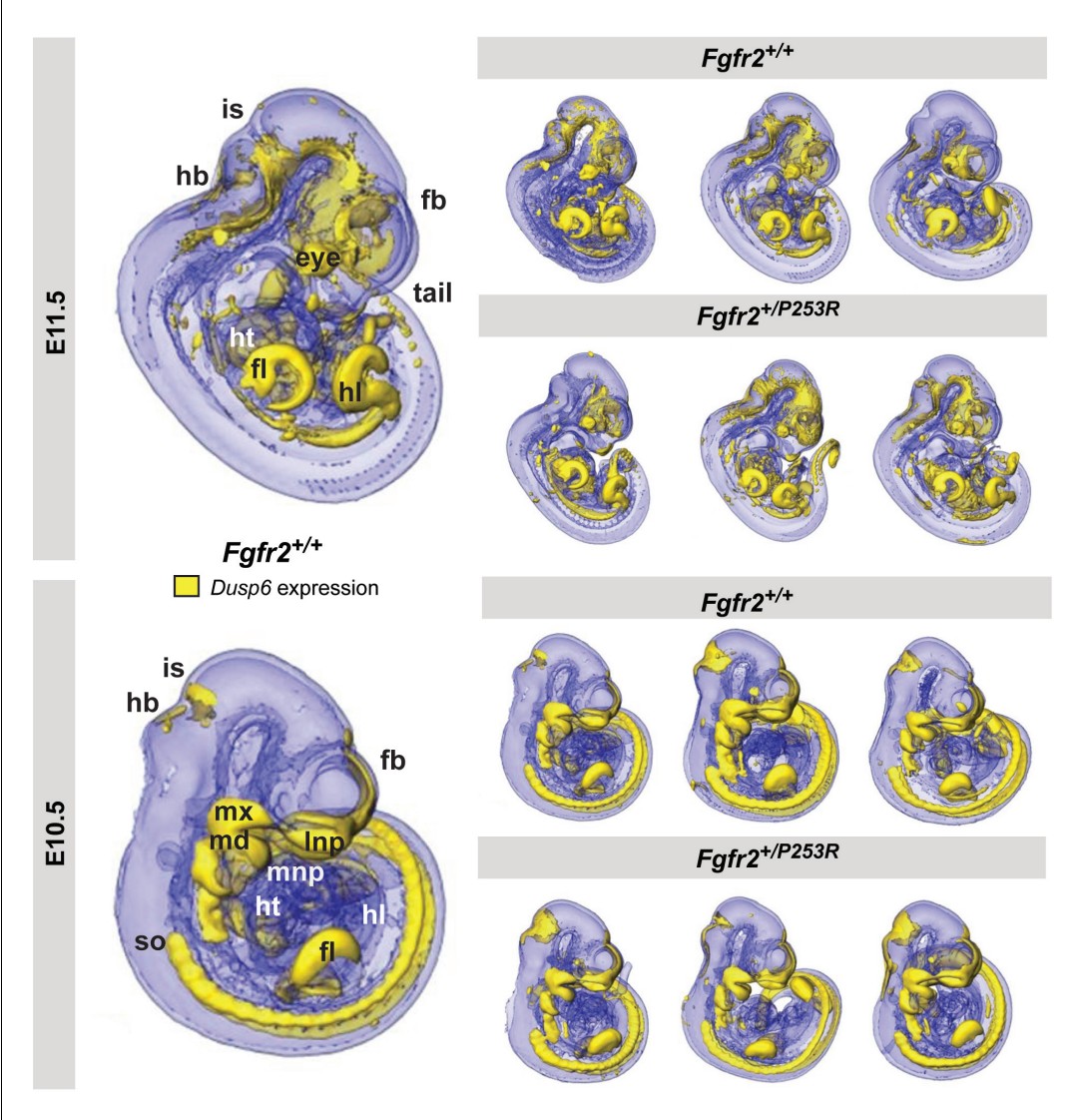

**Figure 2.** Qualitative visualization of *Dusp6* gene expression in unaffected and *Fgfr2+/P253R* mouse embryos at E10.5 and E11.5. OPT scans of embryos WMISH stained for *Dusp6* revealed the anatomical location of *Dusp6* gene expression (shown in yellow). For each stage, the main expression domains are highlighted on the left for anatomical reference on a lateral view of a 3D reconstruction of a *Fgfr2+/+* unaffected embryo (fb, forebrain; fl, forelimb; hb, hindbrain; hl, hindlimb; ht, heart; is, isthmus; lnp, lateral nasal process; md, mandibular prominence; mnp, medial nasal process; mx, maxillary prominence; so, somites). On the right, 3D reconstructions of three unaffected and three *Fgfr2+/P253R* mutant embryos from the same litter are displayed to represent the high degree of variation in developmental age within litters. Embryos are not to scale. Original 2D images of the *Dusp6* WMISH experiments are available in *Figure 2—source data 1*.

DOI: https://doi.org/10.7554/eLife.36405.007

The following source data is available for figure 2:

**Source data 1.** Original 2D images of the *Dusp6* WMISH experiments.
DOI: https://doi.org/10.7554/eLife.36405.008

## The first signs of limb dysmorphology in Apert syndrome

As gene expression patterns are highly dynamic and rapidly change in size, shape and position within a few hours as development progresses (*Martínez-Abadías et al., 2016*), individual limb buds from *Fgfr2+/P253R* mutant embryos and their unaffected littermates aged between E10.5 and E11.5 were staged using a fine-resolution staging system (https://limbstaging.embl.es) (*Musy et al., 2018*). The staging results showed that the analyzed limbs represent a temporal continuum

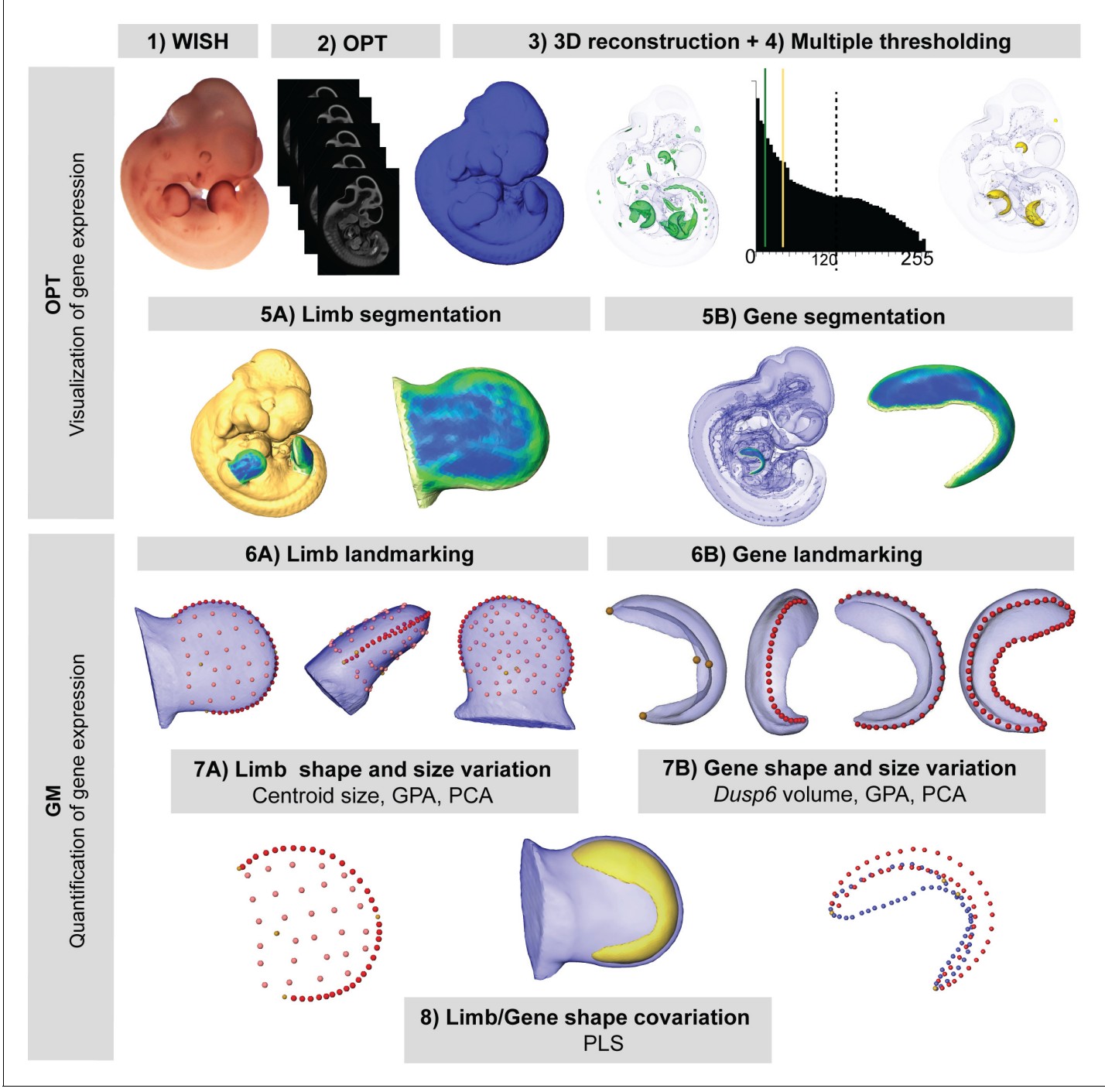

**Figure 3.** New quantitative analysis method for 3D gene expression data, based on Geometric Morphometrics. Mouse embryos between E10.5 and E11.5 were analyzed with WMISH to reveal the expression of *Dusp6* (1), and then cleared with BABB, and OPT scanned using both fluorescence and transmission light (2). The external surface of the embryo was obtained from the 3D reconstruction of the fluorescence scan (2). Multiple thresholding of the transmission scan by choosing different levels of grey values as shown by the histogram allowed the visualization of gene expression patterns at different intensities (3). Moderate gene expression (shown in green) was displayed as the isosurface obtained using a threshold of the grey value computed as 2/3 of the last grey value showing the *Dusp6* expression domain (3). High gene expression (shown in yellow) was displayed as the isosurface obtained using a threshold of the grey value computed as 1/3 of the last grey value showing the *Dusp6* expression domain (3). From the whole mouse embryo isosurfaces, all four limb buds were segmented (5A). From the high gene expression isosurface, the *Dusp6* domains from all of the available limbs were segmented (5B). Maximum curvature patterns were displayed to optimize landmark recording (5). For each limb, we captured the shape and size of the limb bud (6A) and the underlying high *Dusp6* gene expression pattern (6B), recording the 3D coordinates of anatomical landmarks (yellow dots), curve semi-landmarks (red dots) and surface semi-landmarks (pink dots). Anatomical and curve landmarks were recorded

*Figure 3 continued on next page*

*Figure 3 continued*

manually on each limb. Surface landmarks were recorded on one template limb and interpolated onto target limbs (*Video 1*). Landmark coordinates were the input for Geometric Morphometric (GM) quantitative shape analysis (7, 8) that was used to superimpose the landmark data (GPA, Generalized Procrustes Analysis), to compute limb size (centroid size), and to explore both shape variation within limbs and gene expression domains within litters by Principal Component Analysis (PCA). Finally, the covariation patterns for the shape of the limb and the shape of the gene expression domain were also explored using the Partial Least Squares (PLS) method.

DOI: https://doi.org/10.7554/eLife.36405.009

throughout development, with no significant differences between the staging of unaffected and mutant littermates of the same litter (*Figure 4*). We partitioned the time span from E10 to E11.5 into four periods, each one approximately representing 12 hr of development (see *Table 2* and 'Materials and methods' for further details on sample composition).

We first focused on analyzing limb dysmorphology, aiming to determine the youngest stage at which there were morphological differences between mutant and unaffected limbs. To find the earliest moment when differences arise, we traced limb development backwards in time using Geometric Morphometric methods, starting first with embryos from the oldest period (as the differences would be easier to find) and from there proceeding towards the earlier (younger) periods. In this way, we should be able to identify confidently the initiation of limb dysmorphogenesis associated with the *Fgfr2 P253R* mutation.

During the 'Late' period, we detected that *Fgfr2^{+/P253R}* limb buds were already clearly separated from those of their unaffected littermates in the morphospace defined by the Principal Component Analysis (PCA) (*Figure 5A*, *Figure 5—figure supplement 1*). Relative to those of their unaffected littermates, the limbs of *Fgfr2^{+/P253R}* mice presented subtle phenotypic limb differences: limbs were shorter and thicker, with limited development of the wrist (*Figure 5A*). Quantitative comparison of limb size showed that the limbs of mutant mice were also significantly smaller than those of their unaffected littermates (*Table 3* and *Figure 6A*). Interestingly, in this 'Late' period the limbs of mutant mice were smaller than unaffected littermates (although this was not the case in earlier periods). Overall, these results confirmed that the *Fgfr2 P253R* Apert syndrome mutation has an effect on limb development, altering both the size and shape of the limbs. These subtle but significant phenotypic differences would most probably have remained undetected by a qualitative approach. Our quantitative approach revealed their statistical significance and pointed to the origin of the Apert syndrome limb malformation prior to E11.5, before the 'Late' period.

During the 'Mid late' period, the limbs of *Fgfr2^{+/P253R}* mutant mice were still distinguishable from those of unaffected mice in the morphospace of the PCA (*Figure 5B*). At this period, the limbs of *Fgfr2^{+/P253R}* mice lacked the antero-posterior asymmetry and the narrowing of the wrist region more typical of unaffected littermates. Instead, *Fgfr2^{+/P253R}* mice showed a limb phenotype that was elongated in the proximo-distal axis and thickened in the dorso-ventral

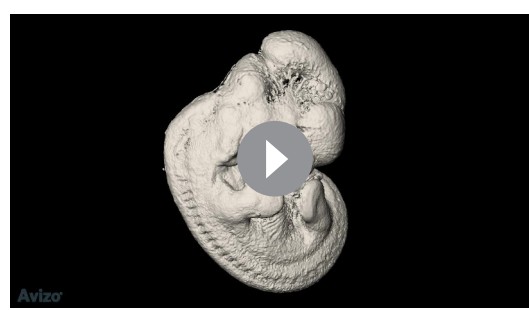

**Video 1.** Morphometric approach for the analysis of 3D gene expression data obtained by Optical Projection Tomography. In this video, we show a superimposed view of a fluorescent and a transmission OPT scan of the same E11.5 mouse embryo, which was analyzed with WMISH to reveal the gene expression of *Dusp6*, a downstream target of Fgf signaling. The surface of the mouse embryo rotating around its central axis was obtained from the 3D reconstruction of an OPT scan based on fluorescence light. The expression pattern of the *Dusp6* gene was segmented from the transmission OPT scan using multiple thresholding. The imaging revealed the anatomical location of regions of high *Dusp6* gene expression (shown in purple). To compare unaffected and *Fgfr2^{+/P253R}* mouse embryos and to detect statistically significant differences between them, we collected landmarks along the surface of the gene expression domain (yellow dots) and the limb bud (green dots), which capture the size and shape of the limb and the underlying gene expression domain to a highly accurate level of detail. Further quantitative analyses provided insight into the origins of limb malformations in Apert syndrome.

DOI: https://doi.org/10.7554/eLife.36405.010

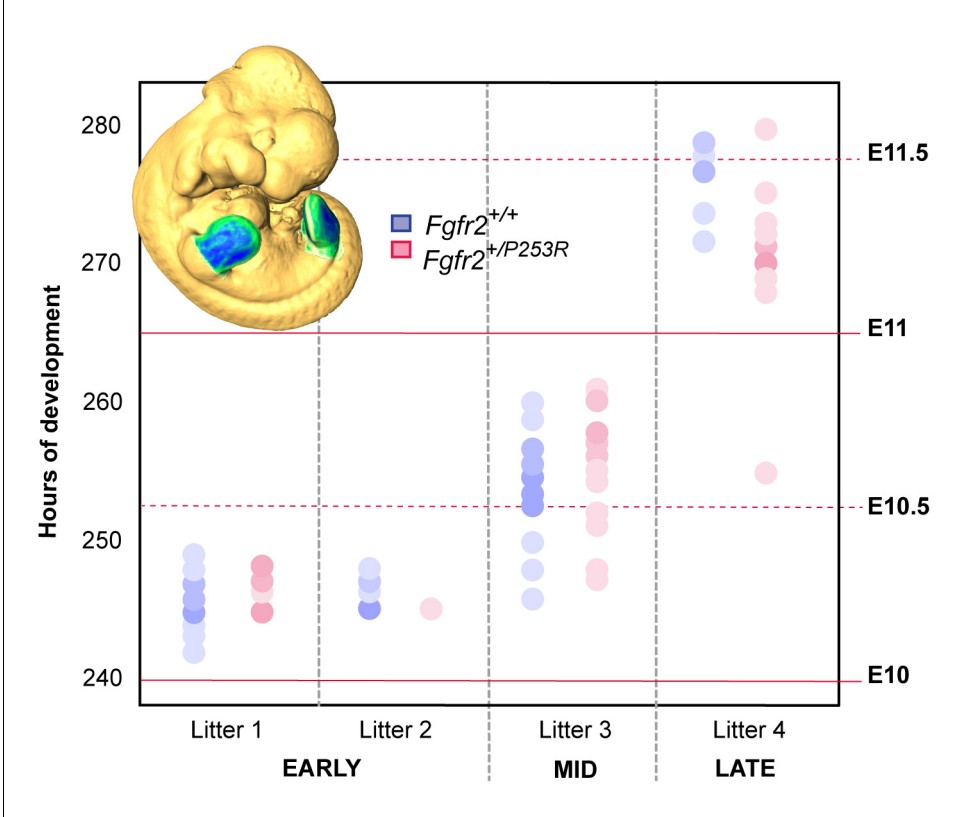

**Figure 4.** Developmental variation within litters of the Apert syndrome mouse model. All of the limbs were individually staged using our publicly available web-based staging system (https://limbstaging.embl.es). The stage of the limb bud was estimated after alignment and shape comparison of the spline with an existing dataset with a reproducibility of ±2 hr. As both Litter 1 and Litter 2 comprised limbs between E10 and E10.5, we pooled these limbs into the 'Early' period. Litter 3, comprising limbs between E10 and E11, represented the 'Mid' developmental period. And finally, Litter 4, which included limbs between E11 and E11.5, was considered as the 'Late' period. In subsequent analyses, forelimbs and hindlimbs were analyzed separately, as specified in *Table 2*. Increasing color intensities in dots represent an increasing number of limbs with the same staging result. Individual scores are available in *Figure 4—source data 1*.

DOI: https://doi.org/10.7554/eLife.36405.011

The following source data is available for figure 4:

**Source data 1.** Staging results.

DOI: https://doi.org/10.7554/eLife.36405.012

axis (*Figure 5B*), resembling the limb shape of younger unaffected embryos. This shape difference coincided with reduced growth in *Fgfr2*$^{+/P253R}$ mice, as the limbs of *Fgfr2*$^{+/P253R}$ mice tended to be smaller than unaffected limbs (*Table 3* and *Figure 6A*). Therefore, significant differences between unaffected and mutant limbs could still be detected during the 'Mid late' period of development and the origins of limb defects associated with Apert syndrome should be sought earlier in development.

The first period during which no significant differences could be detected between unaffected and *Fgfr2*$^{+/P253R}$ Apert syndrome mice was at the 'Mid early' period (*Figure 5C*). Despite internal variation in limb shape, with *Fgfr2*$^{+/P253R}$ mice spreading throughout the morphospace and unaffected littermates concentrated on one region, mutant and unaffected littermates completely overlapped. Therefore, limb shape differences could no longer be detected between groups. Limb size differences were not significant either (*Table 3* and *Figure 6A*). Therefore, our results suggest that the critical time point of limb dysmorphogenesis associated with Apert syndrome occurred between

**Table 2.** Sample composition by genotype and period.

*Fgfr2+/+*: unaffected littermates; *Fgfr2+/P253R*: Apert syndrome mutant littermates. Developmental periods were defined according to limb staging, as shown in **Figure 4**. Hindlimbs and forelimbs were analyzed separately because, in the same embryo, hindlimbs are delayed in development in comparison to forelimbs by about 6–8 hr. We thus presented in the main text the results from groupings highlighted in color, including those for the hindlimbs from the early and mid groups, as well as those for the forelimbs from the mid and the late groups. This represents a continuous time span of limb development, from E10 to E11.5, divided into four periods of 12 hr each. Results from the complete dataset that considers hindlimbs and forelimbs from each litter separately are available in **Figures 5** and **6** (**Figure 5—figure supplement 2**, **Figure 5—figure supplement 3**, **Figure 6—figure supplement 1** and **Figure 6—figure supplement 2**).

| Genotype | Period | N limb | N gene | LIMB | | | |
|---|---|---|---|---|---|---|---|
| | | | | Hindlimb (N = 34) | | Forelimb (N = 46) | |
| | | | | Early (~E10) | | | |
| | | | | Limb | Gene | Limb | Gene |
| *Fgfr2+/+* | EARLY: E10-E10.5 | 30 | 24 | 13 | 9 | 17 | 15 |
| *Fgfr2+/P253R* | | 16 | 15 | 11 | 9 | 5 | 6 |
| Subtotal | | 46 | 39 | 24 | 18 | 22 | 21 |
| | | | | Mid early (~E10.5) | | Mid late (~E11) | |
| | | | | Limb | Gene | Limb | Gene |
| *Fgfr2+/+* | MID: E10.5-E11 | 15 | 10 | 7 | 3 | 8 | 7 |
| *Fgfr2+/P253R* | | 16 | 9 | 8 | 3 | 8 | 6 |
| Subtotal | | 31 | 19 | 15 | 6 | 16 | 13 |
| | | | | | | Late (~E11.5) | |
| | | | | Limb | Gene | Limb | Gene |
| *Fgfr2+/+* | LATE: E11-E11.5 | 8 | 8 | 4 | 4 | 4 | 4 |
| *Fgfr2+/P253R* | | 15 | 14 | 7 | 6 | 8 | 8 |
| *Subtotal* | | 23 | 22 | 11 | 10 | 12 | 12 |
| Total | | 100 | 80 | | | | |

DOI: https://doi.org/10.7554/eLife.36405.013

the 'Mid late' and 'Mid early' periods, corresponding to the transition period from E10.5 to E11 (**Figure 5B–C**).

Consistent with the above, no further sign of limb shape dysmorphology was detected during the 'Early' period of development (**Figure 5D**). During this period, there was a great range of developmental variation, with unaffected and *Fgfr2+/P253R* mutant mice completely overlapping in the morphospace and with all limbs displaying similar incipient bud shapes (**Figure 5D**). The limbs of *Fgfr2+/P253R* mice were significantly larger than the limbs of their unaffected littermates (**Table 3** and **Figure 6A**), suggesting that at this early time point, there is a significant effect of the *Fgfr2 P253R* mutation on limb size but not on limb shape (**Figure 6A**).

### *Fgfr2* Apert syndrome mutation leads to aberrant overexpression of *Dusp6* domains

We next sought to obtain direct evidence of altered genetic regulation that could explain the observed limb phenotype by analyzing the shape dynamics of the expression of *Dusp6*, a direct target gene of Fgf signaling. As in the limb shape analysis described above, we chose to work backwards in developmental time, first examining the gene expression of *Dusp6* in the embryos from the latest period. We found that in the 'Late' period, *Dusp6* expression was already different in *Fgfr2+/P253R* mutant mice and unaffected littermates. The differences were significant both in shape (**Figure 5E**) and size (**Table 3** and **Figure 6B**). In the limbs of unaffected mice, the *Dusp6* expression domain appeared as a thin domain underlying the apical ectodermal ridge, whereas in *Fgfr2+/P253R* mutant mice, the shape of the *Dusp6* domain was expanded in all directions (**Figure 5E**). Accordingly, the volume of the *Dusp6* expression domain was significantly larger in Apert syndrome mice

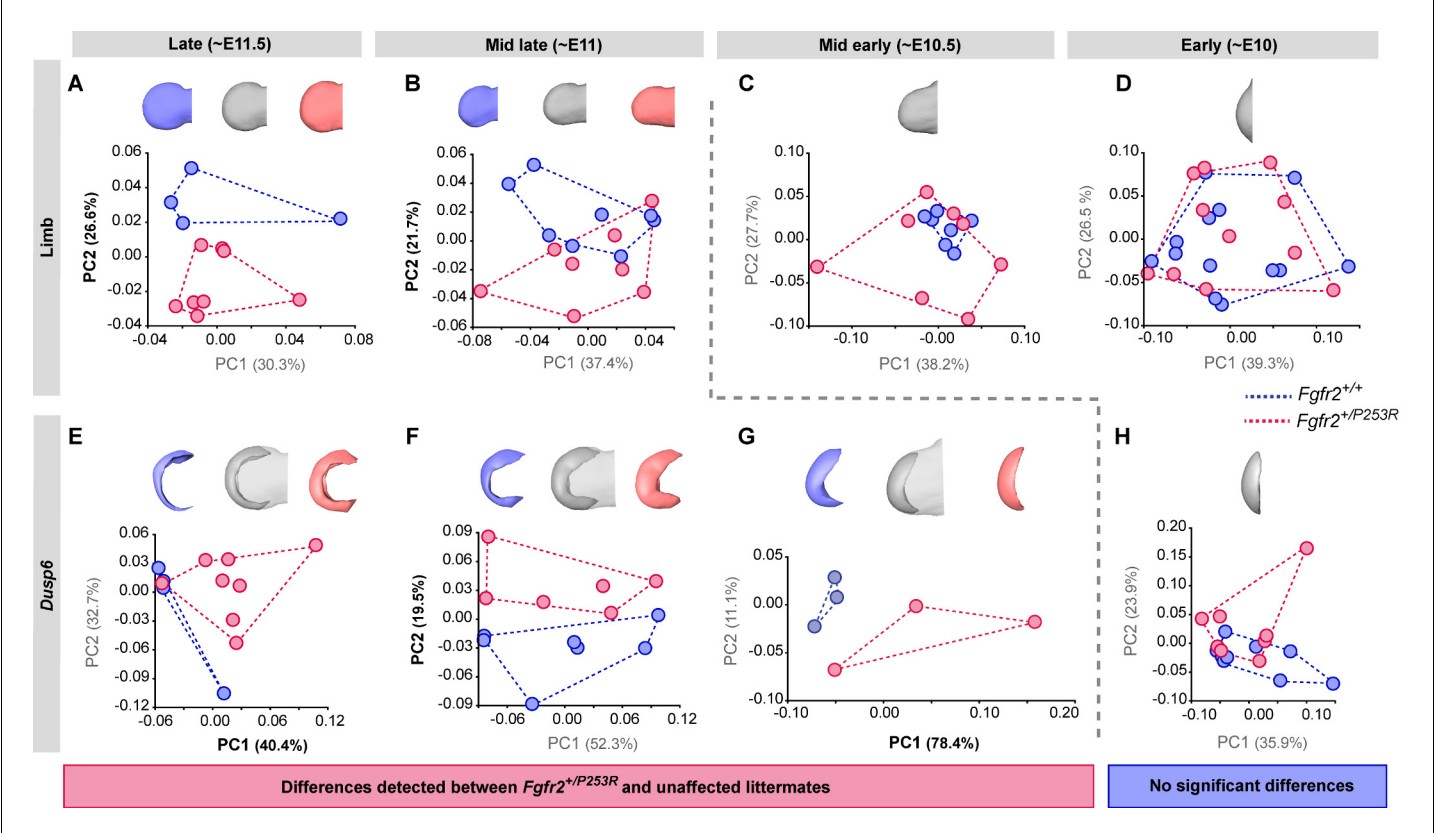

**Figure 5.** Tracing of limb phenotypes (anatomical and molecular) back through developmental time to the earliest moment of appearance. Principal Component Analysis based on the Procrustes-based semi-landmark analysis was used to assess the shape of the limbs and the corresponding *Dusp6* expression domains for each developmental period (*Figure 5—figure supplement 1*). Each period was analyzed separately for the shape of the limb (A–D) and the *Dusp6* expression domain (E–H), as specified in the 'Materials and methods' and in *Table 2*. Scatterplots of PC1 and PC2 axes with the corresponding percentages of total morphological variation explained are displayed for each analysis, along with the morphings associated with the negative, mid and positive values of the PC axis that separates mutant and unaffected littermates (PC1 or PC2, as highlighted in bold black letters in the corresponding axis). Morphings are displayed in grey tones when the analysis showed no differentiation between mutant and unaffected littermates. Morphings are displayed in color when the analysis revealed differentiation between mouse groups (blue: unaffected littermates; pink: mutant littermates). Limb buds and *Dusp6* domains are not to scale and are oriented with the distal aspect to the left, the proximal aspect to the right, the anterior aspect at the top and the posterior aspect at the bottom of all images. Convex hulls represent the ranges of variation within each group of mice. Individual scores are available in *Figure 5—source data 1*. Results from the complete dataset analyzing hindlimbs (*Figure 5—figure supplement 2*, *Figure 5—figure supplement 2—source data 1*) and forelimbs (*Figure 5—figure supplement 3*, *Figure 5—figure supplement 3—source data 1*) from each litter separately are also available.

DOI: https://doi.org/10.7554/eLife.36405.014

The following source data and figure supplements are available for figure 5:

**Source data 1.** Source files for limb shape data.
DOI: https://doi.org/10.7554/eLife.36405.020

**Figure supplement 1.** Anatomical landmarks recorded on the 3D reconstructions of the limbs and the *Dusp6* expression domains obtained from OPT scans of Apert syndrome embryos at E10.5–E11.5.
DOI: https://doi.org/10.7554/eLife.36405.015

**Figure supplement 2.** Principal Component analyses based on the Procrustes-based semi-landmark analysis of the shape of the hindlimbs and the corresponding *Dusp6* expression domains for each developmental period.
DOI: https://doi.org/10.7554/eLife.36405.016

**Figure supplement 2—source data 1.** Source files for hindlimb shape data.
DOI: https://doi.org/10.7554/eLife.36405.017

**Figure supplement 3.** Principal Component analyses based on the Procrustes-based semi-landmark analysis of the shape of the forelimbs and the corresponding *Dusp6* expression domains for each developmental period.
DOI: https://doi.org/10.7554/eLife.36405.018

**Figure supplement 3—source data 1.** Source files for forelimb shape data.

*Figure 5 continued on next page*

*Figure 5 continued*

DOI: https://doi.org/10.7554/eLife.36405.019

(*Table 3* and *Figure 6B*), even when these mice had significantly smaller limbs (*Figure 6A*). The *Dusp6* expression domain thus grew disproportionately in the limbs of *Fgfr2*$^{+/P253R}$ mutant mice in the latest period of development (*Figure 5E*), which is consistent with reports of whole-body size reduction in *Fgfr2*$^{+/P253R}$ Apert mice and of the over-activation of *Fgfr2* signaling by the Apert syndrome mutation (*Yu and Ornitz, 2001*; *Ibrahimi et al., 2001*).

During the 'Mid late' period, the shapes of expression patterns were still distinct — the expanded *Dusp6* expression domain persisted on the dorsal and ventral sides of mutant limbs, but was reduced on the anterior and posterior sides (*Figure 5F*). The overall volume of the *Dusp6* expression domains remained larger in *Fgfr2*$^{+/P253R}$ mutant mice, but the difference was no longer statistically significant (*Table 3* and *Figure 6B*).

During the 'Mid early' period, the separation between unaffected and *Fgfr2*$^{+/P253R}$ mutant mice was maintained in the PCA analysis (*Figure 5G*). Unaffected mice showed a *Dusp6* expression domain that was expanded towards the anterior and posterior edges of the expression domain (*Figure 5G*). By contrast, *Fgfr2*$^{+/P253R}$ mutant mice did not show the extension and the posterior asymmetry of the *Dusp6* expression domain typical of normal limb development, suggesting a lack of differentiation in the *Dusp6* expression of mutant limbs (*Figure 5G*).

The 'Early' period was the only time point when we did not detect a significant difference between unaffected and *Fgfr2*$^{+/P253R}$ mice (*Figure 5H*). The PCA showed variation in the expression domains of *Dusp6*, with similar gene expression patterns in terms of both shape (*Figure 5H*) and size (*Table 3* and *Figure 6B*) across all mice. Therefore, the first observation of an alteration in the gene expression pattern (*Figure 5G*) occurred earlier than the change in the limb shape (*Figure 5B*). Our analyses provide evidence that differences in the *Dusp6* gene expression pattern first occurred during the 'Mid early period', preceding the phenotypic limb differentiation, which occurred a few hours later during the 'Mid late period'. Overall, the time courses showing the dynamics of limb shape and gene expression pattern changes throughout development (*Figure 5* and *Figure 6*) confirmed that the *Fgfr2* Apert syndrome mutation causes an aberrant overexpression of *Dusp6* early in development, which could later lead to significant limb malformations.

## Altered *Dusp6* expression and limb dysmorphology are highly associated

Finally, we explored the patterns of correlation between the limb phenotype and gene expression to further explore how altered Fgf signaling relates to the limb malformations induced by the Apert syndrome *Fgfr2 P253R* mutation. First, we assessed the relationship between the size of the limbs and the volume of the *Dusp6* expression domain, pooling all the forelimbs and hindlimbs and assessing the correlation between these two traits (*Figure 6C,D*). The trend line showed that for the same limb size, *Fgfr2*$^{+/P253R}$ mutant mice showed larger *Dusp6* expression domains, both in forelimbs ($R^2 = 0.4$) and hindlimbs ($R^2 = 0.6$). If the extension of the *Dusp6* expression depended only on limb growth, and the gene domain passively became larger by just following limb tissue growth, a higher correlation between the size of the limb and the gene expression would be expected. However, the moderate correlation found here suggests that the size of the *Dusp6* gene expression zone is not solely dependent on limb growth but might be controlled by further genetic regulatory factors.

Second, we assessed the morphological integration between the shape of the limbs and the shape of the *Dusp6* expression domain. The statistical analysis of the covariance pattern between these shapes can reflect the interaction of the phenotype and the gene expression pattern during limb development. As shown by analysis of additional genes expressed during limb development (*Martínez-Abadías et al., 2016*), even when a gene is expressed within the limb, the shape of the limb and the shape of the gene expression domain are not correlated by definition, and the integration pattern can change from a strong association to no significant correlation within few hours as the limb develops (*Martínez-Abadías et al., 2016*). The dynamics of the integration pattern can identify the key periods during which the expression of a gene is relevant for determining the shape of the limb. If the morphological integration is low, the expression of the gene will not be as relevant

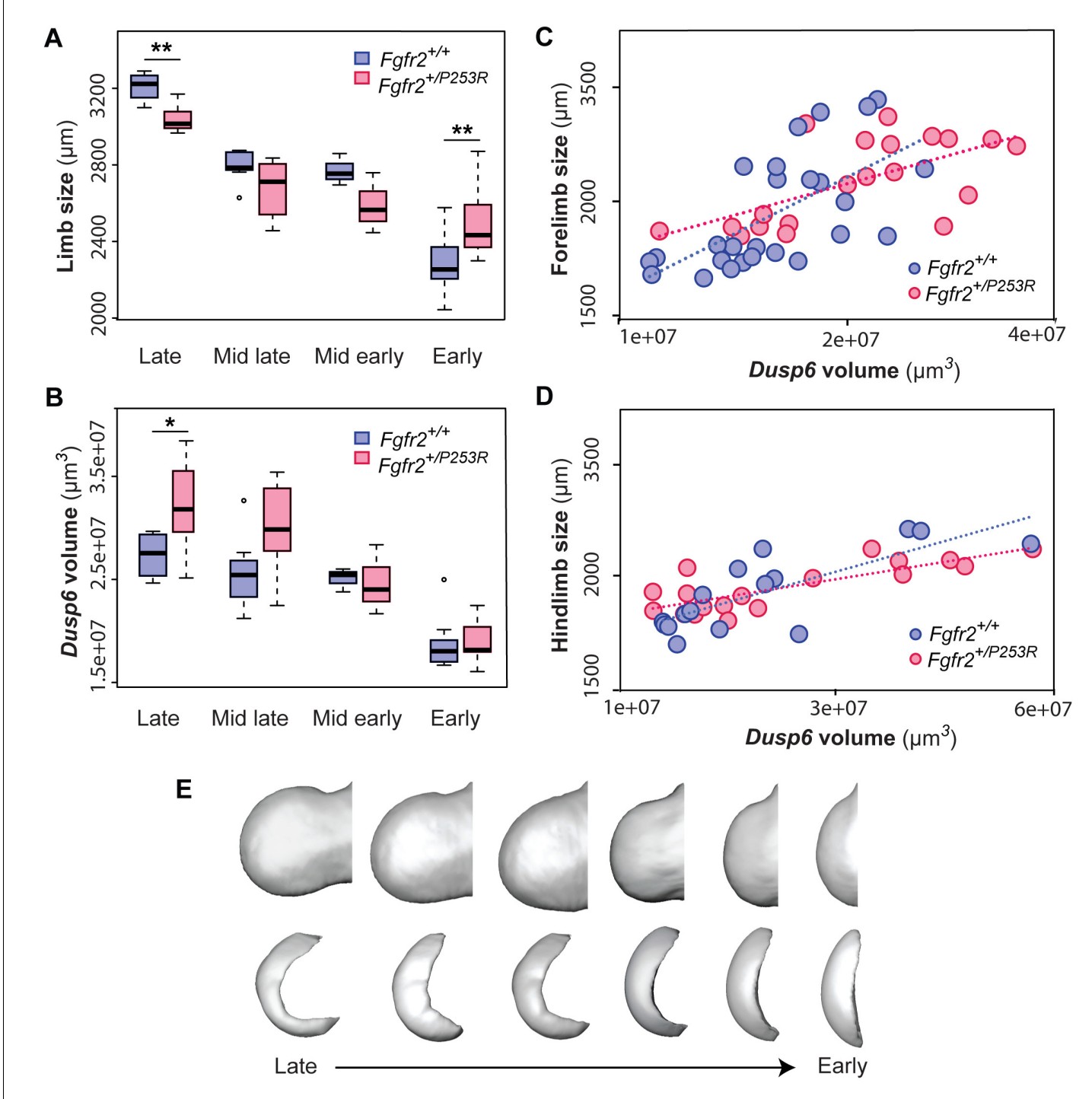

**Figure 6.** Quantitative correlation between the size and shape of the limbs and the *Dusp6* expression pattern. (A–D) Comparison of limb bud size and *Dusp6* volume in unaffected and *Fgfr2*$^{+/P253R}$ mutant littermates across development (*Table 3*). Limb size was measured as limb centroid size (A), whereas the size of the *Dusp6* expression was measured as the volume of the gene domain (B), as specified in the 'Materials and methods' and in *Table 2*. Statistically significant differences as revealed by two-tailed t-tests are marked with asterisks, representing the degree of significance: *P-value*=0.03, **P-value*=0.01. Results from hindlimbs and forelimbs are separately available in *Figure 6—figure supplement 1*. The association between the size of the limbs and the volume of the *Dusp6* domain was assessed separately for forelimbs (C) and hindlimbs (D). Individual scores for all of these analyses are available in *Figure 6—source data 1*. (E) Time course assessing the morphological integration pattern between the limb phenotype and the shape of the gene expression pattern using partial least squares analyses. Associated shape changes from late to early limb development are shown from morphings associated with the negative, mid and positive values of PLS1, which accounted for almost all of the covariation (97.6% in

*Figure 6 continued on next page*

*Figure 6 continued*

forelimbs and 99.5% in hindlimbs) between the limb buds and the *Dusp6* gene expression domains (**Figure 6—figure supplement 2**). Limb buds and *Dusp6* domains are not shown to scale and are oriented distally to the left, proximally to the right, anteriorly to the top and posteriorly to the bottom. On the right, representing the positive extreme of PLS1 axis, typical early limb buds showed a protruding shape (i.e. short in the proximo-distal axis and symmetrical in the antero-posterior axis) associated with a flat-bean shaped *Dusp6* expression zone localized in the distal limb region, underlying the apical ectodermal ridge and spreading proximally towards the dorsal and ventral sides of the limb. On the left, representing the negative extreme of the PLS1 axis, limb buds were elongated in the proximal axis and asymmetric on the antero-posterior axis, with an expansion of the distal limb region and a contraction of the proximal region, at the wrist level. This limb shape, which is typical of more developed limbs, was associated with a *Dusp6* expression that was extended underneath the apical ectodermal ridge towards the anterior and the posterior ends of the gene expression zone, but reduced on the dorsal and ventral sides of the limb. Individual scores for all of these analyses are available in **Figure 6—figure supplement 2—source data 1**. Legends for supplementary figures.

DOI: https://doi.org/10.7554/eLife.36405.021

The following source data and figure supplements are available for figure 6:

**Source data 1.** Source files for limb size and gene expression volume.

DOI: https://doi.org/10.7554/eLife.36405.025

**Figure supplement 1.** Quantitative size analyses of limb bud and *Dusp6* volume in unaffected and *Fgfr2^{+/P253R}* mutant littermates throughout development.

DOI: https://doi.org/10.7554/eLife.36405.022

**Figure supplement 2.** Morphological integration between the shapes of the limb bud and the *Dusp6* gene expression domain, as measured by partial least square (PLS) analysis.

DOI: https://doi.org/10.7554/eLife.36405.023

**Figure supplement 2—source data 1.** Results of Ppartial Least Squares (PLS) analyses.

DOI: https://doi.org/10.7554/eLife.36405.024

for determining the shape of the limb as it would be when the integration is high and the impact of the altered gene expression on the phenotype is minimal. If the morphological integration is high, the impact of the genetic mutation will be maximized (i.e. changes in the gene expression pattern will produce changes in the limb shape). Our results showed that the shape of the limb and the shape of the *Dusp6* expression domain were indeed highly correlated (RV = 0.88 in forelimbs; RV = 0.91 in hindlimbs). This is evidence that altered Fgf signaling, induced by the *Fgfr2 P253R* Apert syndrome mutation, can be associated with limb dysmorphology.

By comparing the morphological integration patterns in mutant and unaffected littermates during different periods, we can test whether this interaction is maintained or disrupted by the disease during development. If the pattern or magnitude of morphological integration is different in mutant mice, it should reveal further mechanisms underlying the etiology of the disease. Our analyses showed that the pattern of morphological integration of the shape of the limbs and the shape of the *Dusp6* expression domains was similar in unaffected and *Fgfr2^{+/P253R}* mutant mice (**Figure 6E**,

**Table 3.** Quantitative comparison of limb size (µm) and *Dusp6* volume (µm$^3$) in *Fgfr2^{+/P253R}* mutant and *Fgfr2^{+/+}* unaffected mice. Results from two-sided t-tests are provided separately for forelimbs and hindlimbs and for each developmental group, as defined in **Table 2** and **Figure 4**. Statistically significant differences are marked with * (*P-value*<0.05).

| | | EARLY | | | | | MID | | | | | LATE | | | | |
|---|---|---|---|---|---|---|---|---|---|---|---|---|---|---|---|---|
| | | *Fgfr2^{+/+}* | *Fgfr2^{+/P253R}* | t | df | P-value | *Fgfr2^{+/+}* | *Fgfr2^{+/P253R}* | t | df | P-value | *Fgfr2^{+/+}* | *Fgfr2^{+/P253R}* | t | df | P-value |
| Forelimb | Limb size | 2041.63 | 2274.15 | 6.99 | 18.98 | <0.0001 | 2728.24 | 2579.69 | −1.70 | 7.74 | 0.13 | 3311.66 | 3097.72 | −3.57 | 5.39 | 0.01 |
| | *Dusp6* volume | 18160720 | 18387239 | 0.13 | 13.03 | 0.89 | 23002206 | 27432909 | 1.78 | 9.64 | 0.11 | 24792568 | 29875554 | 2.47 | 9.72 | 0.03 |

| | | EARLY | | | | | MID | | | | | LATE | | | | |
|---|---|---|---|---|---|---|---|---|---|---|---|---|---|---|---|---|
| | | *Fgfr2^{+/+}* | *Fgfr2^{+/P253R}* | t | df | P-value | *Fgfr2^{+/+}* | *Fgfr2^{+/P253R}* | t | df | P-value | *Fgfr2^{+/+}* | *Fgfr2^{+/P253R}* | t | df | P-value |
| Hindlimb | Limb size | 2280.36 | 2489.62 | 2.68 | 15.76 | 0.02 | 2769.51 | 2589.95 | −1.74 | 3.03 | 0.18 | 3191.12 | 2955.91 | −3.44 | 6.73 | 0.01 |
| | *Dusp6* volume | 16294103 | 16277958 | −0.01 | 15.36 | 0.99 | 23920390 | 23131303 | −0.19 | 2.45 | 0.86 | 41802827 | 45608477 | 0.52 | 4.14 | 0.63 |

DOI: https://doi.org/10.7554/eLife.36405.026

*Figure 6—figure supplement 2*). Our results confirmed that the *Fgfr2 P253R* mutation does not disrupt the strong association between limb shape and the *Dusp6* expression domain. Therefore, the alteration of the *Dusp6* expression pattern between E10 and E11.5, caused by the *Fgfr2* mutation, will correlate with the limb dysmorphologies associated with Apert syndrome. Overall, the high correlation between the shape of the limb and the *Dusp6* expression domain provides further evidence that altered Fgf expression resulting from the *Fgfr2* mutation is strongly associated with limb defects in Apert syndrome.

## Discussion

By definition, revealing the primary etiology of an abnormality requires going back in time to the earliest moment when abnormal development can be found. Typically, the earliest changes will be the most subtle, and so the most statistically sensitive techniques are necessary to reveal these changes. The methods that are currently used to assess gene expression patterns are mainly qualitative and focus only on shape and size differences that can be simply detected by eye. Therefore, slight changes in gene expression domains, even if they have large effects on the phenotype (*Honeycutt, 2008*), can remain undetected. To reveal these subtle changes, we have developed a precise method combining OPT and GM to quantify embryo morphology and the underlying 3D gene expression patterns in a systematic, objective manner. This enables the visualization and quantification of how the genotype translates into the phenotype during embryonic development, the comparison of normal and disease-altered patterns of genetic and phenotypic variation and, eventually, the identification of the origins of abnormal morphogenesis. This approach can further our understanding of the etiology of genetic diseases in research using animal models (*Spradling et al., 2006*; *Rosenthal and Brown, 2007*), even in models that do not seem to recapitulate the human disease faithfully (*Guénet, 2011*).

Our study of the *Fgfr2 P253R* mouse model for Apert syndrome is an exemplary case demonstrating how quantitative assessment can overcome the shortcomings of traditional qualitative morphological assessment and lead to new discoveries. To date, the molecular and developmental mechanisms underlying the limb defects associated with Apert syndrome have remained obscure, even when these limb abnormalities clinically differentiate Apert syndrome from other craniosynostosis syndromes (such as Pfeiffer, Crouzon and Saethre-Chotzen syndromes) (*Park et al., 1995*; *Holten et al., 1997*; *Cohen and MacLean, 2000*). Most Apert syndrome research has focused on premature fusion of cranial sutures and craniofacial malformations (*Cohen and MacLean, 2000*; *Holmes et al., 2009*; *Martínez-Abadías et al., 2010*; *Holmes and Basilico, 2012*; *Hill et al., 2013*; *Heuzé et al., 2014*), clinical traits that are consistently phenocopied in mouse models. However, little research has been carried out on the limb defects in Apert syndrome using the same animal models, mainly because most previous research reported that malformations of the limbs are subtle or absent in the different mouse models for Apert syndrome (*Chen et al., 2003*; *Wang et al., 2005*; *Wang et al., 2010*). Contrary to these previous results, our quantitative morphometric analyses demonstrate that the limbs of *Fgfr2$^{+/P253R}$* Apert syndrome mice have significant defects that are detectable in newborn mice and that can be traced back to early embryogenesis (*Figure 1*, *Figure 5* and *Figure 6*).

Our analyses provide insight into the genetic origins of these limb defects, showing that altered expression patterns of genes in the Fgf signaling pathway precede and contribute to limb dysmorphogenesis in *Fgfr2$^{+/P253R}$* Apert syndrome mice. In fact, we detected that *Dusp6* expression patterns were different in unaffected and mutant littermates a few hours before the first limb dysmorphologies appeared (*Figure 5*), and confirmed that limb shape and *Dusp6* expression patterns were highly correlated (*Figure 6E*, *Figure 6—figure supplement 2*). The altered Fgf signaling that was observed was due to the *Fgfr2* P253R Apert syndrome mutation, which causes loss of ligand specificity of the receptor and increased affinity of ligands for Fgfr2. As the Apert syndrome mutation is located in the linker region between the second and the third Ig-like domains of the receptor, and as alternative splicing of the protein occurs in the carboxy-terminal half of IgIII domain, the mutation affects both isoforms of the receptor, Fgfr2IIIb and Fgfr2IIIc (*Ibrahimi et al., 2001*). Fgfr2IIIb is the isoform that is predominantly expressed in epithelial cells which normally binds to ligands produced in mesenchymal cells, whereas Fgfr2IIIc is expressed in mesenchymal cells and binds to ligands produced in epithelial cells. The available evidence indicates that Apert mutations

alter this ligand specificity and thus raise the level of Fgfr2 signaling — either through the aberrant interaction of mutant receptors with inappropriate ligands or through the upregulation of Fgfr2 (*Hajihosseini, 2008*) and unpublished data). Our analyses showed that over-activation of the Fgf signaling pathway results in more expanded (*Figure 5E–G*) and larger (*Figure 6B*) expression domains for *Dusp6*, a gene that acts as a negative-feedback control of Fgf signaling (*Ekerot et al., 2008*). A delay in misregulation of the expression of *Dusp6* may explain the lack of antero-posterior asymmetry and the shape deficiencies observed in *Fgfr2$^{+/P253R}$* mutant mice (*Figure 5*). We found evidence that limb shape and *Dusp6* expression were highly associated (*Figure 6C–E*), but it is likely that other downstream genes of the Fgf signaling pathway also contribute to the limb shape malformations associated with Apert syndrome.

Our analyses also demonstrated that the embryonic limb defects persisted until birth (*Figure 1*) and thus did not disappear during development. For instance, we found that *Fgfr2$^{+/P253R}$* Apert syndrome mice presented postnatal limb malformations involving the shape, length and volume of many bones of the forelimb, including the scapula, humerus, ulna, radius, metacarpals and phalanges (*Table 1* and *Figure 1*). It is not possible, however, to correlate early changes in limb bud shape directly to the defects in chondrogenesis and long bone length that appear later in development. The skeletal defects associated with Apert syndrome are the result of continuous altered Fgf signaling throughout growth and development. Overall, activated Fgf signaling may lead to decreased bone growth (*Li et al., 2007*). Our results demonstrate that *Dusp6* participates in this process, beginning very early in development, as do many other downstream targets of Fgf signaling that have different functions and spatio-temporal patterns, all of which are involved in the complex process that regulates limb morphogenesis. Although subtle, the first malformations detected in this study suggest that further research into the origins and causes of limb dysmorphologies in Apert syndrome using these and other mouse models is warranted.

Our quantitative approach could be similarly applied to investigate other developmental defects and dysmorphologies (*Winter and Baraitser, 1987*). This is relevant as major developmental defects represent a leading cause of infant mortality and affect a small but relevant percentage of the population, severely compromising their quality of life (*Toxicology NRC C on D, 2000*). OPT and GM can potentially be used to analyze any organ and animal model that can be visualized during development using light microscopy, and any gene whose gene expression pattern can be detected by WMISH and shows a continuous expression domain over development (*Martínez-Abadías et al., 2016*). Our study is a proof of concept that demonstrates that this approach is feasible and develops a protocol that could be adjusted for the study of different genes and organs. With the current pipeline, the implementation of the approach should be straightforward and no more time- or resource-consuming than other cellular or molecular experiments. WMISH labeling is a standard technique in most molecular and developmental laboratories. OPT imaging is increasingly accessible and software for performing GM is freely available.

We exemplified the method by analyzing limb defects in mouse models, but it could be applied to malformations affecting the face, the brain, the tail, or any other organ that involves complex 3D shapes that are quantifiable with GM. Organs with morphologies that are devoid of reliable anatomical landmarks, such as the developing heart, should be analyzed with alternative non-landmark-based methods, such as spherical harmonics (*Shen et al., 2009*). Moreover, the OPT-GM approach could be applied to other vertebrate animal models, such as zebrafish, chicken and *Xenopus*. Finally, our approach could also be extended to analyze protein expression patterns. Immunostaining labeling with a secondary antibody coupled to a BAAB-resistant fluorophore, such as Alexa, will reveal the expression pattern of proteins and this information can be precisely captured and quantified in 3D using the OPT-GM approach.

The limitation of the approach is the requirement to generate large samples of wildtype and mutant mice per gene and time point, since several WMISH labeling and OPT scanning activities cannot be performed within the same embryos. For high-throughput analyses assessing the expression of multiple genes, such as transcriptomic and microarray assays (*Yeh et al., 2013*), our OPT-GM approach is a complementary technique that could confirm whether the differences in the level of gene expression of candidate genes are associated with changes in the patterns of this expression (i.e. the regions within a tissue where the genes are expressed) and further phenotypic malformations. This type of quantitative analyses will lead to a deeper understanding of how development translates genetic into phenotypic variation.

Through its increased quantitative sensitivity, our method has allowed us to reveal that the mouse model for Apert syndrome does indeed show very early abnormalities in limb development. We detected that dysregulation of an *Fgfr2* target gene precedes measurable changes in limb bud morphology, thus identifying a relevant component of its genetic etiology. Quantitative evaluation of size and shape should thus be performed before discarding any animal models as useful for investigating human congenital malformations (*Zuniga et al., 2012*). Our method has the potential to become a useful tool for biomedical research, providing insight into the processes that cause malformations and lead to malfunction, which is essential for understanding diseases and discovering potential therapies.

# Materials and methods

## Key resources table

| Reagent type (species) or resource | Designation | Source or reference | Identifiers | Additional information |
|---|---|---|---|---|
| Genetic reagent (*M. musculus*) | *Fgfr2*$^{+/P253R}$ | *Wang et al., 2010*; doi: 10.1186/1471-213X-10–22 | | Laboratory of Dr. Richtsmeier (Pennsylvania State University); inbred mouse model on a C57BL/6J background |
| Chemical compound, drug | PBST | Sigma-Aldrich | P3563 | Phosphate-buffered saline, 0.1% tween 20 |
| Chemical compound, drug | paraformaldehyde | Sigma-Aldrich | P6148 | 4% in PBS |
| Chemical compound, drug | methanol | Sigma-Aldrich | 494437–2L-D | Methanol for protein sequencing, bioReagent, 99.93% |
| Chemical compound, drug | digoxigenin-UTP | Sigma-Aldrich | 11277073910 | DIG RNA Labeling Mix (Roche) |
| Sequence-based reagent | *Dusp6* | *Dickinson et al. (2002)* doi:10.1016/S0925-4773 (02)00024–2 | | m Mkp3-pCVM.sport6 |
| Antibody | anti-DIG-AP (sheep polyclonal) | Sigma-Aldrich | 11093274910 | (1:2000) |
| Other | NBT | Sigma-Aldrich | 11585029001 | 4-nitro blue tetrazolium chloride, crystals (Roche) |
| Other | BCIP | Sigma-Aldrich | 11383221001 | 4-toluidine salt (Roche) |
| Chemical compound, drug | BABB | Sigma-Aldrich | 402834; W213802 | (one benzyl alcohol: two benzyl benzoate) |
| Chemical compound, drug | agarose | Sigma-Aldrich | A9414 | Agarose low gelling temperature |
| Software, algorithm | Matlab | The MathWorks, Inc | RRID:SCR_001622 | https://es.mathworks.com/products/matlab.html |
| Software, algorithm | R, CRAN | *R Development Core Team, 2014* | RRID:SCR_003005 | http://www.R-project.org/ |
| Software, algorithm | Amira 6.3 | FEI | RRID:SCR_014305 | https://www.fei.com/software/amira-3d-for-life-sciences/ |
| Software, algorithm | Viewbox | dHAL software, Kifissia, Greece | RRID:SCR_016481 | http://www.dhal.com/ |
| Software, algorithm | Geomorph | *Adams and Otárola-Castillo, 2013*; doi: 10.1111/2041-210X.12035 | RRID:SCR_016482 | https://cran.r-project.org/web/packages/geomorph/index.html |

*Continued on next page*

*Continued*

| Reagent type (species) or resource | Designation | Source or reference | Identifiers | Additional information |
|---|---|---|---|---|
| Software, algorithm | MorphoJ | *Klingenberg (2011)*: doi: 10.1111/j.1755 –0998.2010.02924.x | RRID:SCR_016483 | http://www.flywings.org. uk/morphoj_page.htm |

## Mouse model

We analyzed the $Fgfr2^{+/P253R}$ Apert syndrome mouse model, an inbred model backcrossed on the C57BL/6J genetic background for more than ten generations, which carries a mutation that in humans with Apert syndrome is associated with more severe syndactyly. This gain-of-function mutation, which is embryonically lethal in homozygosis, involves a proline to arginine amino acid change at position 253 of the $Fgfr2$ protein, which alters the ligand-binding specificity of the receptor and causes stronger receptor signaling. Further details of the mouse model and on the generation of targeting construct can be found elsewhere (*Wang et al., 2010*). All the experiments were performed in compliance with the animal welfare guidelines approved by the Pennsylvania State University Animal Care and Use Committees (IACUC46558, IBC46590).

## Micro-CT imaging

High-resolution micro-computed tomography (µCT) scans were acquired by the Center for Quantitative Imaging at the Pennsylvania State University (www.cqi.psu.edu) using the HD-600 OMNI-X high-resolution X-ray computed tomography system (Varian Medical Systems, Palo Alto, CA). Pixel sizes ranged from 0.01487 to 0.01503 mm, and all slice thicknesses were 0.016 mm. Image data were reconstructed on a 1024 × 1024 pixel grid as a 16-bit TIFF. To reconstruct forelimb morphology from the µCT images, isosurfaces were produced with median image filter using the software package Avizo 6.3 (Visualization Sciences Group, VSG) (*Figure 1A–F*).

## Morphometrics in P0 mice

We assessed forelimb morphology at P0 using unaffected (N = 10) and mutant (N = 12) littermates of Apert syndrome mouse models. A selection of 54 landmarks was collected on the left forelimb of all P0 mice, as shown in *Figure 1B–F*, following anatomical criteria selected to best describe the size and shape of forelimb bones in a reliable and reproducible manner. Landmarks at the proximal and distal tips of the phalanges, metacarpals, radius, ulna and clavicle were collected to represent the simple tubular shape of these bones. Additional landmarks were collected in anatomical structures of the humerus and scapula to better represent their complex 3D shapes (*Figure 1B–F*). To minimize measurement error, each landmark was collected twice by the same observer, restricting the deviations between the two trials to 0.05 mm.

At P0, we estimated the dimensions of the long bones of the forelimbs using the 3D coordinates of the landmarks located at the proximal and distal ends of the bones (*Figure 1B–F*). We also estimated the bone volumes from volume data collected from the microCT scans. To assess size differences in bone length and bone volume between mutant and unaffected P0 mice of the $Fgfr2^{+/P253R}$ Apert syndrome mouse model, we performed a two-tailed one-way ANOVA on those variables showing a normal distribution, and the non-parametric Mann-Whitney U-Test on those variables that deviated from a normal distribution. The shape of the humerus and the scapula was comparatively assessed in unaffected and $Fgfr2^{+/P253R}$ Apert syndrome littermates using Geometric Morphometrics.

Shape information was extracted using a Generalized Procrustes Analysis (GPA) (*Rohlf and Slice, 1990*), in which configurations of landmarks are superimposed by shifting them to a common position, rotating and scaling them to a standard size until a best fit of corresponding landmarks is achieved (*Dryden and Mardia, 1998*). The resulting Procrustes coordinates from the GPA were the input for further statistical analysis to compare the shape of the bones in unaffected and $Fgfr2^{+/P253R}$ mice. A Principal Component Analysis (PCA) was used to explore the morphological variation within each bone. PCA performs an orthogonal decomposition of the data and transforms variance covariance matrices into a smaller number of uncorrelated variables called Principal Components (PCs),

which successively account for the largest amount of variation in the data (*Hallgrimsson et al., 2015*). Each specimen is scored for every Principal Component and the specimens can be plotted using these scores along the morphospace defined by the principal axes.

## WMISH and OPT scanning

To examine early embryonic mouse limb development in Apert syndrome mice, we bred four litters of the *Fgfr2$^{+/P253R}$* Apert syndrome mouse model and collected them between E10.5 and E11.5. In total, 32 mouse embryos were harvested and classified by PCR genotyping into unaffected (N = 16) and mutant (N = 16) littermates (see *Table 2* and *Figure 4* for further details on sample size and composition). We analyzed the maximum number of samples that we could process simultaneously within the same WMISH batch, as explained next.

*Dusp6* gene expression was assessed by whole-mount-in-situ hybridization (WMISH). Mouse embryos were dissected in cold phosphate-buffered saline, 0.1% tween 20 (PBST), fixed overnight in 4% paraformaldehyde (Sigma), dehydrated in a graded PBST/methanol series and stored at –20°C in methanol. The mouse embryos recovered their original size after rehydration in decreasing series of methanol/PBST. WMISH was carried out using *Dusp6* antisense RNA probes labeled with digoxigenin-UTP (Roche), following standard protocols (*de la Pompa et al., 1997*). Alkaline phosphatase coupled anti-digoxigenin (anti-DIG-AP, Roche) and NBT/BCIP staining (Roche) were used to reveal the expression pattern for *Dusp6*. To minimize variations during experimental procedures, all embryos were processed systematically within the same batch, processing unaffected and mutant littermates from different litters in separate tubes, but simultaneously using the same probe, timings and concentration reagents.

After embedding in agarose, dehydrating in methanol and chemically clearing the samples with benzyl alcohol and benzyl benzoate (BABB), the stained whole embryos were scanned with both fluorescence and transmission light with a cyan fluorescent protein (CFP) filter using our home-built OPT imaging system mounted on a Leica MZ 16 FA microscope (*Sharpe et al., 2002*). The embryos were 3D-reconstructed from the resulting 2D images using Matlab (The MathWorks, Inc.) and visualized using Amira 6.3 (Visualization Sciences Group, FEI). From the OPT fluorescence scans, we produced 3D reconstructions of the embryo surface and we dissected the available right and left fore- and hindlimbs of each specimen, resulting in a sample of 100 embryonic limbs (*Table 2*). From the OPT transmission scans, we recovered the *Dusp6* expression domain. As *Dusp6* is expressed in a fuzzy spatial gradient, as already shown by other genes (*Martínez-Abadías et al., 2016*), we used 3D multiple thresholding to visualize the gene expression domain at different intensities (*Figure 3*). To analyze the gene expression domains comparatively across the samples, we inspected the whole range of threshold values under which the gene expression could be visualized for each limb, from the threshold showing its first appearance to the threshold under which it disappeared and was no longer detectable. We analyzed the 3D reconstruction on the basis of a threshold computed as 1/3 that of the last grey value showing the *Dusp6* expression domain, which displayed a *Dusp6* domain at high gene expression (*Figure 3*, step 3). Finally, we obtained 80 limbs (46 forelimbs and 34 hindlimbs) with associated gene expression patterns for *Dusp6*. Limbs or gene expression domains that were damaged during experimental manipulation or which showed obvious artifacts resulting from altered staining or imaging were discarded from the analyses. Outlier points were detected following standard protocols in GM that are based on squared Procrustes distance, and landmark misplacements were fixed whenever possible.

## Embryo staging

To account for breeding and developmental variation, individual limb buds were staged using our publicly available web-based staging system (https://limbstaging.embl.es) (*Musy et al., 2018*). Considering the spline curve along the outline of the limb, this tool provides a stage estimate with a reproducibility of ±2 hr. According to the staging results, the different mouse litters were ordered following a continuous temporal sequence from E10 to E11.5 (*Table 2* and *Figure 4*). To minimize high developmental variation within and among litters of mice, hindlimbs and forelimbs from each litter were analyzed separately, except for those from two litters from the earliest stage that were pooled into the same group because their temporal distribution completely overlapped, as shown in *Figure 4*.

## Morphometrics from E10 to E11.5

To capture the size and shape of the limbs and the expression domains of *Dusp6,* we collected a set of anatomical landmarks as well as curve and surface semi-landmarks (*Figure 5—figure supplement 1*), as recommended in structures devoid of homologous landmarks. Semi-landmarks are mathematical points located along a curve (*Bookstein, 1997*) or a surface (*Gunz et al., 2005*) within the same object that can be slid to corresponding equally spaced locations across the sample. Only five anatomical landmarks were discernible in the limb, and four in the *Dusp6* expression pattern (*Figure 5— figure supplement 1*). We defined several curves and surfaces over the limb as well as gene expression structures with a relatively low number of semi-landmarks in order to minimize the high dimensionality of the data while providing an adequate shape coverage that precisely captured the subtle morphological changes associated with the Apert syndrome mutation. We used Amira 6.3 (Visualization Sciences Group, FEI) to record the anatomical landmarks and Viewbox 4 (dHAL software, Kifissia, Greece) to construct a limb template of surface and curve semi-landmarks and to interpolate them onto each target shape (*Figure 5—figure supplement 1*).

The 3D landmark coordinates defining the shape of the limb and the *Dusp6* expression domain were analyzed using Procrustes-based landmark analysis (*Bookstein, 1997*). Semi-landmarks were allowed to slide in the GPA by minimizing the bending energy (*Bookstein, 1997*; *Gunz et al., 2005*; *Mitteroecker and Gunz, 2009*). Quantitative shape analyses based on PCA were performed as explained above.

We estimated the size of the limb as the centroid size, calculated as the square root of the summed squared distances between each landmark coordinate and the centroid of the limb configuration of landmarks (*Dryden and Mardia, 1998*). The volumes of the *Dusp6* domains were estimated from the 3D reconstructions of the OPT scans. Differences in limb size and gene expression volume between mutant and unaffected embryonic mice were tested for statistical significance using a two-tailed Welch Two Sample t-test.

We quantified the integration between the limb and the *Dusp6* expression pattern and produced visualizations of the patterns of associated shape changes between them using two-block Partial Least Squares analysis (PLS) (*Rohlf and Corti, 2000*). This method performs a singular value decomposition of the covariance matrix between the two blocks of shape data (i.e., the limb and the *Dusp6* expression domain). Uncorrelated pairs of new axes are derived as linear combinations of the original variables, with the first pair accounting for the largest amount of inter-block covariation, the second pair for the next largest amount and so on. The amount of covariation is measured by the RV coefficient, which is a multivariate analogue of the squared correlation (*Klingenberg, 2009*). Statistical significance was tested using permutation tests under the null hypothesis of complete independence between the two blocks of variables. Separate analyses for each developmental period, as well as for forelimbs and hindlimbs of all stages, were computed.

All of the analyses were performed using R (*R Development Core Team, 2014*; http://www.R-project.org); the R package geomorph (*Adams and Otárola-Castillo, 2013*; http://cran.r-project.org/web/packages/geomorph), SPSS Statistics 22 (IMB, 2013) and MorphoJ (*Klingenberg, 2011*).

## Acknowledgements

We acknowledge support of the Spanish Ministry of Economy and Competitiveness provided to the EMBL partnership and the 'Centro de Excelencia Severo Ochoa', as well as support from the CERCA Programme/Generalitat de Catalunya. We thank Ethylin Wang Jabs for access to the *Fgfr2*[+/P253R] Apert syndrome mouse model. The research leading to these results received funding from the following grants: a European Union Seventh Framework Program (FP7/2007-2013) under grant agreement Marie Curie Fellowship FP7-PEOPLE-2012-IIF 327382, National Institutes of Health grants NICHD P01HD078233 and NIDCR R01DE02298, and a Burroughs-Welcome Fund 2013 Collaborative Research Travel Grant.

# Additional information

## Funding

| Funder | Grant reference number | Author |
|---|---|---|
| European Commission | FP7-PEOPLE-2012- 597 IIF 327382 | Neus Martínez-Abadías |
| National Institutes of Health | NICHD P01HD078233 | Joan Richtsmeier |
| National Institutes of Health | NIDCR R01DE02298 | Joan Richtsmeier |
| Burroughs Wellcome Fund | 2013 Collaborative Research Travel Grant | Joan Richtsmeier |

The funders had no role in study design, data collection and interpretation, or the decision to submit the work for publication.

## Author contributions

Neus Martínez-Abadías, Conceptualization, Data curation, Formal analysis, Supervision, Funding acquisition, Investigation, Methodology, Writing—original draft, Project administration; Roger Mateu Estivill, Jaume Sastre Tomas, Formal analysis, Methodology, Writing—review and editing; Susan Motch Perrine, Data curation, Formal analysis, Supervision, Methodology, Writing—review and editing; Melissa Yoon, Formal analysis, Methodology; Alexandre Robert-Moreno, Jim Swoger, Visualization, Methodology, Writing—review and editing; Lucia Russo, Visualization, Methodology; Kazuhiko Kawasaki, Methodology, Writing—review and editing; Joan Richtsmeier, Conceptualization, Resources, Data curation, Supervision, Funding acquisition, Project administration, Writing—review and editing; James Sharpe, Conceptualization, Supervision, Funding acquisition, Project administration, Writing—review and editing

## Author ORCIDs

Neus Martínez-Abadías (iD) https://orcid.org/0000-0003-3061-2123
Roger Mateu Estivill (iD) http://orcid.org/0000-0003-4728-2239
Jaume Sastre Tomas (iD) https://orcid.org/0000-0001-9053-2390
Alexandre Robert-Moreno (iD) https://orcid.org/0000-0001-9042-1316
Jim Swoger (iD) http://orcid.org/0000-0003-3805-0073
James Sharpe (iD) https://orcid.org/0000-0002-1434-9743

## Ethics

Animal experimentation: All the experiments were performed in compliance with the animal welfare guidelines approved by the Pennsylvania State University Animal Care and Use Committees (IACUC46558, IBC46590).

## Decision letter and Author response

Decision letter https://doi.org/10.7554/eLife.36405.031
Author response https://doi.org/10.7554/eLife.36405.032

# Additional files

## Supplementary files

• Transparent reporting form
DOI: https://doi.org/10.7554/eLife.36405.027

## Data availability

Our dataset has been deposited to Dryad (http://dx.doi.org/10.5061/dryad.8h646s0).

The following dataset was generated:

**Database, license,**

| Author(s) | Year | Dataset title | Dataset URL | and accessibility information |
|---|---|---|---|---|
| Martinez-Abadias N, Estivill RM, Tomas JS, Perrine SM, Yoon M, Robert-Moreno A, Swoger J, Russo L, Kawasaki K, Richtsmeier J, Sharpe J | 2018 | Data from: Quantification of gene expression patterns to reveal the origins of abnormal morphogenesis | http://dx.doi.org/10.5061/dryad.8h646s0 | Available at Dryad Digital Repository under a CC0 Public Domain Dedication |

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
