## [Decision Letter]

Thank you for submitting your article "Quantification of gene expression patterns to reveal the origins of abnormal morphogenesis" for consideration by *eLife*. Your article has been reviewed by Marianne Bronner as the Senior Editor, a Reviewing Editor, and three reviewers. The following individuals involved in review of your submission have agreed to reveal their identity: Nele A Haelterman (Reviewer #1); Karen Lyons (Reviewer #2); Susan Brooks (Reviewer #3).

The reviewers have discussed the reviews with one another and the Reviewing Editor has drafted this decision to help you prepare a revised submission.

The reviewers and editors were enthusiastic about the optical projection techniques coupled with gene expression that should provide new insights with other mutant strains, although there was some question about how frequently this methodology could be employed in other laboratories. The method was clearly able to identify morphological anomalies earlier in development than previously recognized in the context of the *Fgfr2* mutation and the relationship of those changes to *Dusp6* expression. There were however some concerns which need to be discussed in the revision. In particular these were related to *Dusp6* functional data and the correlation with limb size. Other comments are noted below and should be addressed.

*Reviewer #1:*

Summary:

Revealing the pathogenic mechanism that causes a developmental disorder typically requires phenotypic characterization of model organisms that are mutant for the homolog of the causative human gene. When such mutation induces a subtle phenotype, it is quite challenging to identify the changes, making it practically impossible to tease out the underlying pathogenic mechanisms.

In the current manuscript, Martinez-Abadias et al. combine multiple methods (whole mount in situ hybridization, optical projection tomography and geometric morphometrics) to quantify subtle changes in the morphology and gene expression of *Fgfr2^+/P253R^* mutant limb buds. Whereas previous studies failed to identify significant changes in limb development in these mutants, the method developed by Martinez-Abadias et al. allowed for a more in-depth analysis, revealing subtle, but significant changes between the limbs of *Fgfr2^+/P253R^* mutants and their wildtype littermates. The authors next describe similar subtle, but significant changes in the expression pattern of a downstream target of the *FGFR2: Dusp6*.

This is a well-written manuscript that presents a novel method to quantify gene expression patterns in 3D. In my opinion, it provides a significant advance to the field, as it is not simple to quantify expression patterns and pinpoint changes in 3D, even if these changes are large.

Essential revisions:

1) This is an interesting new technology to identify modest changes in morphology of a developing organ or body part. However, I am not convinced it is feasible to perform these experiments at a higher throughput to analyze the expression pattern of multiple genes and be able to truly dissect out which genes have altered expression patterns in a mutant. The method appears to be quite labor intensive. Does one need to limit themselves to analyzing one or two candidate genes whose expression is already thought to be altered? Currently, the authors show a proof of concept of one gene. Could the authors comment on the time it took or would take to analyze the expression pattern of a novel gene, assuming the WISH is already optimized?

2) The authors reveal an expansion of the *Dusp6* expression domain in developing mutant limb buds and show a high correlation between the shape of the limb and the shape of the *Dusp6* domain. However, based on the presented data, the statement that "This is evidence that altered *Fgf* signaling induced by the *Fgfr2* P253R Apert syndrome mutation will have a direct effect on the limb phenotype” (subsection “Altered *Dusp6* expression and limb dysmorphology are highly associated”) is an overstatement. The authors show a correlation, but to definitively show that altered *Dusp6* expression and/or *Fgf* signaling is causative, functional data is required. Does normalizing the *Dusp6* expression domain, or normalizing *FGFR2* function in these mutant mice rescue the observed phenotypes? If the authors have such data, it should be added to the manuscript. If not, this statement should be toned down in subsection “Altered *Dusp6* expression and limb dysmorphology are highly associated” and elsewhere in the manuscript.

*Reviewer #2:*

This manuscript describes the use of optical projection tomography (OPT) and geometric morphometrics (GM) to address one of the biggest challenges in relating genotype to phenotypic outcome. I am not an expert on OPT or GM, and cannot evaluate the technical details in this manuscript. However, this group has done foundational studies in this area, described the methodology in detail, and has provided all source data. This will enable other groups to replicate the analysis and compare it with other methods in the future.

There are other studies using both GM and OPT from this group and others. Experts in these techniques will need to weigh in on the technical novelty in this manuscript. The conceptual novelty here is to apply these methodologies to quantify and changes in gene expression and relate these to changes in form. The analysis assesses variation within a genotype and relates this to variation in stages of development. This is powerful. The analysis also firmly establishes differences in gene expression and form based on *FGFR2*genotype. Neither of these insights are possible using standard histological or morphological analysis, and from my limited knowledge, is a serious challenge even for experts in quantitative 3D analysis. Therefore, this report is an important demonstration. The results demonstrate that effects on *Dusp6* expression can be seen in limb buds of *Fgfr2 P253R* mice. This is an interesting and important finding. It is also highly plausible. It opens the door to finding how the expression domain of *FGFR2* in normal and mutant limb buds intersects with that of *Dusp6*. It is less clear how, or if, these early changes in limb bud shape relate to the defects in chondrogenesis and long bone length, since *FGFR2* is continuously expressed in skeletal tissues.

The data are extensive and clearly presented, even to this reviewer, who does not have expertise with the types of analysis that are employed. Some of the data in the figures are replicated in the supplemental figures, but this is helpful to follow the logic. While the typical laboratory would not be able to establish this methodology, the study could inspire very insightful collaborations, and should inform the larger publicly supported efforts to catalog both gene expression and mutant phenotypes.

In summary, the main value of the study lies in its promise as a methodology to quantify gene expression changes and relate these to subtle morphological outcomes.

*Reviewer #3:*

The manuscript describes an application of a new methodology correlating spatially mapped quantitative gene expression data with shape changes to examine the development of skeletal abnormalities. Specifically, mice possessing a mutation in the *Fgfr2* gene were analyzed for forelimb development and expression of *Dusp6*, a target of *Fgf* signaling. The method was able to identify morphological anomalies earlier in development than previously recognized in the context of the *Fgfr2* mutation and the relationship of those changes to *Dusp6* expression.

The paper is well-written and the data are clearly presented. Clarification of some methods and additional discussion of a few points would improve the manuscript.

How were the landmarks on the forelimb bones chosen? Are these defined elsewhere or did the authors choose them for this work specifically? In the latter case, what provided the rationale for the choices? Similarly, please describe how the markers for the limb and gene landmarking were determined.

Subsection “Altered *Dusp6* expression and limb dysmorphology are highly associated”. The authors characterize the correlations between *Dusp6* expression and limb size as moderate, but the R2 values indicate that ~50% of the variation in limb size is explained by *Dusp6* expression. This actually seems quite large for a single gene. In the same discussion, the authors describe *Dusp6* gene expression's dependency on limb size. Is the directionality of the relationship from size to *Dusp6* expression or is it changes in *Dusp6* expression that partially determine limb size?

Discussion section. The authors indicated that the approach could be similarly applied to other developmental defects in other organ systems. A discussion of which tissues or organs are particular amenable to this method or may be particularly problematic would be an interesting addition.

---

## [Author Response]

The reviewers and editors were enthusiastic about the optical projection techniques coupled with gene expression that should provide new insights with other mutant strains, although there was some question about how frequently this methodology could be employed in other laboratories.

We really appreciate all the positive feedback from the editors and reviewers and we are glad to share their enthusiasm. We acknowledge that the potential applications of our method in other laboratories, working with other genes or organs, was not sufficiently explained in the previous version of the manuscript. In this review, we have expanded the Discussion section to address this issue and provide more examples in which our method could be directly applied to address questions of interest to a wide range of researchers.

The method was clearly able to identify morphological anomalies earlier in development than previously recognized in the context of the Fgfr2 mutation and the relationship of those changes to Dusp6 expression. There were however some concerns which need to be discussed in the revision. In particular these were related to Dusp6 functional data and the correlation with limb size. Other comments are noted below and should be addressed.

See response to each of these issues below and corresponding changes in the main text (Results section).

Reviewer #1:[…]Essential revisions:1) This is an interesting new technology to identify modest changes in morphology of a developing organ or body part. However, I am not convinced it is feasible to perform these experiments at a higher throughput to analyze the expression pattern of multiple genes and be able to truly dissect out which genes have altered expression patterns in a mutant.

Our study was a proof of concept to demonstrate that this approach was feasible and to establish a basic protocol that could later be adjusted to different genes and organs. We agree with the reviewer that once we have demonstrated the potential of the method, the next step would be to leverage the method to perform higher throughput analyses. The current method should be readily extended to other organs, genes and/or species provided that their shapes are well-defined with at least some reliably recognizable landmarks, and the genes can be labeled by WMISH (Martínez-Abadías et al., 2016).

The bottleneck is, however, the generation of so many mouse embryos to achieve large wildtype and mutant mouse samples per gene and time point. Unfortunately, it is technically not possible to perform several WMISH labeling and OPT scanning within the same embryos. Therefore, rather than using the OPT-GM method as an exploratory technique to analyze the expression pattern of myriad of genes, it is more plausible to use the OPT-GM as a method to confirm whether the expression patterns of candidate genes that have been identified using other molecular techniques, such as transcriptomics, are altered and can be associated with phenotypic malformations. This type of quantitative analyses is the only way that will lead to a deeper understanding of how development translates genetic into phenotypic variation. This knowledge will be essential in the future for understanding diseases and discovering potential therapies that can be introduced just before the first alterations appear. This issue is further discussed now in the main text of the manuscript. Please, see the Discussion section.

The method appears to be quite labor intensive. Does one need to limit themselves to analyzing one or two candidate genes whose expression is already thought to be altered? Currently, the authors show a proof of concept of one gene. Could the authors comment on the time it took or would take to analyze the expression pattern of a novel gene, assuming the WISH is already optimized?

The manuscript reflects how labor intensive it was to develop the technique for the first time. Now that the experimentation, imaging, segmentation and landmark processing of the samples have all been optimized, and are explained in full detail in the paper, a similar sample as the one used in this study could be processed in several weeks. The number of genes to be assessed will depend on the intrinsic nature of the investigation and the available resources.

One of the main apparent difficulties to implement the OPT-GM method might be the lack of familiarity with Geometric Morphometrics (GM) within the Developmental Biology field. However, GM is currently used in many laboratories to analyze the patterns of morphological variation in all types of organisms (Klingenberg, 2010). Thanks to the growing interest in this technique, many resources are available to learn and perform GM studies (http://life.bio.sunysb.edu/morph/index.html). Actually, all the analyses in this study have been performed using freely available software and scripts that were rewritten to automatize the process, as for example *Fiji* (Fiji is just ImageJ) and R packages especially suited to GM, such as geomorph.

An extended discussion on these issues can be found at the following link:

https://prelights.biologists.com/highlights/quantification-gene-expression-patterns-reveal-origins-abnormal-morphogenesis/

This issue has been dealt with in the text in the Discussion section.

2) The authors reveal an expansion of the Dusp6 expression domain in developing mutant limb buds and show a high correlation between the shape of the limb and the shape of the Dusp6 domain. However, based on the presented data, the statement that "This is evidence that altered Fgf signaling induced by the Fgfr2 P253R Apert syndrome mutation will have a direct effect on the limb phenotype (subsection “Altered Dusp6 expression and limb dysmorphology are highly associated”) is an overstatement. The authors show a correlation, but to definitively show that altered Dusp6 expression and/or Fgf signaling is causative, functional data is required. Does normalizing the Dusp6 expression domain, or normalizing FGFR2 function in these mutant mice rescue the observed phenotypes? If the authors have such data, it should be added to the manuscript. If not, this statement should be toned down in subsection “Altered Dusp6 expression and limb dysmorphology are highly associated”and elsewhere in the manuscript.

We acknowledge that the sentence in subsection “Altered *Dusp6* expression and limb dysmorphology are highly associated” is an overstatement. Correlation is not causation and functional data is not available, as we have not tested any means to normalize Dups6 expression or *FGFR2* function in these mice. Therefore, we have toned it down accordingly – we now say “This is evidence that altered *Fgf* signaling induced by the *Fgfr2* P253R Apert syndrome mutation can be associated with limb dysmorphology”. Please, see change in subsection “Altered *Dusp6* expression and limb dysmorphology are highly associated”.

Reviewer #2:This manuscript describes the use of optical projection tomography (OPT) and geometric morphometrics (GM) to address one of the biggest challenges in relating genotype to phenotypic outcome. I am not an expert on OPT or GM, and cannot evaluate the technical details in this manuscript. However, this group has done foundational studies in this area, described the methodology in detail, and has provided all source data. This will enable other groups to replicate the analysis and compare it with other methods in the future.There are other studies using both GM and OPT from this group and others. Experts in these techniques will need to weigh in on the technical novelty in this manuscript. The conceptual novelty here is to apply these methodologies to quantify and changes in gene expression and relate these to changes in form. The analysis assesses variation within a genotype and relates this to variation in stages of development. This is powerful. The analysis also firmly establishes differences in gene expression and form based on FGFR2 genotype. Neither of these insights are possible using standard histological or morphological analysis, and from my limited knowledge, is a serious challenge even for experts in quantitative 3D analysis. Therefore, this report is an important demonstration. The results demonstrate that effects on Dusp6 expression can be seen in limb buds of FGFR2 P253R mice. This is an interesting and important finding. It is also highly plausible. It opens the door to finding how the expression domain of FGFR2 in normal and mutant limb buds intersects with that of Dusp6. It is less clear how, or if, these early changes in limb bud shape relate to the defects in chondrogenesis and long bone length, since FGFR2 is continuously expressed in skeletal tissues.

As correctly pointed out by the reviewer, we cannot correlate early changes in limb bud shape to the defects in chondrogenesis and long bone length. Indeed, the skeletal defects associated with Apert syndrome are the result of continuous altered *FGF/FGF*R signaling throughout growth and development. However, this was not the goal of our study – our goal was to determine whether we could find subtle early changes in gene expression (and limb bud shape) before any phenotype could be detected by eye. Our results demonstrate that our method is able to identify the first moment in which limb malformation and altered *Dusp6* expression arises. However, our results do not imply that limb dysmorphologies are caused only by altered expression of *Dusp6* in E10.5. Besides *Dusp6*, many other downstream targets of *Fgf* signaling must be involved in this complex process regulating limb morphogenesis. This has been addressed in the Discussion section.

The data are extensive and clearly presented, even to this reviewer, who does not have expertise with the types of analysis that are employed. Some of the data in the figures are replicated in the supplemental figures, but this is helpful to follow the logic. While the typical laboratory would not be able to establish this methodology, the study could inspire very insightful collaborations, and should inform the larger publically supported efforts to catalog both gene expression and mutant phenotypes.In summary, the main value of the study lies in its promise as a methodology to quantify gene expression changes and relate these to subtle morphological outcomes.

We really appreciate the reviewer’s enthusiastic overall comment about our study.

Reviewer #3:[…]The paper is well-written and the data are clearly presented. Clarification of some methods and additional discussion of a few points would improve the manuscript.How were the landmarks on the forelimb bones chosen? Are these defined elsewhere or did the authors choose them for this work specifically? In the latter case, what provided the rationale for the choices? Similarly, please describe how the markers for the limb and gene landmarking were determined.

The landmarks on the forelimb bones were chosen by us for this work specifically following anatomical criteria to best describe their size and shape in a reliable and reproducible manner. Indeed, one of the main advantages of GM is that each study is unique and can be personalized to meet the requirements of the analysis. The set of landmarks used to capture the shape of the object/organism under study is always chosen by the investigators. The definitions of many anatomical landmarks can be found in the literature, but additional new landmarks can be used by the researchers whenever a clear definition is provided to ensure the reproducibility of the study. Below we provide further explanation about how we chose our set of landmarks.

For the hand, clavicle and forearm bones (radius and ulna), as they present simple tubular shapes, we just collected the landmarks at the proximal and distal tips of the bones, as shown in Figure 1 and explained in the figure legend. From the 3D coordinates of these landmarks, we could estimate total bone length by applying the Pythagorean Theorem. In the case of the humerus, besides the landmarks at the proximal and distal tips, we also collected a landmark at the tip of the deltoid process, which is an additional readily recognizable feature of this bone. Finally, in the case of the scapula, which presents a more complex 3D shape, we captured its morphology by precisely registering the 3D coordinates of points defined by anatomical structures of the scapula, such as the spine, the acromion process and the glenoid cavity.

In GM, this type of homologous anatomical landmarks are the basis of a rigorous quantitative shape analysis. Anatomical landmarks can be combined with the so-called semilandmarks, which are points located along a curve or a surface that can be slid to corresponding equally spaced locations (Gunz, Mitteroecker, and Bookstein, 2005). This is an approach widely applied in structures presenting few homologous landmarks. As only five landmarks were discernable in the limb, and four in the *Dusp6* expression pattern, we defined several curves and surfaces over the limb and gene expression structures to precisely capture the subtle morphological changes associated with the Apert syndrome mutation. The number of semilandmarks is also chosen by the investigators, usually compromising between a good shape coverage and minimizing the high dimensionality of the data. Further details are provided in Figure 5—figure supplement 1. We have added some explanations regarding these topics in Materials and methods subsection “Morphometrics from E10 to E11.5”.

Subsection “Altered Dusp6 expression and limb dysmorphology are highly associated”. The authors characterize the correlations between Dusp6 expression and limb size as moderate, but the R2 values indicate that ~50% of the variation in limb size is explained by Dusp6 expression. This actually seems quite large for a single gene. In the same discussion, the authors describe Dusp6 gene expression's dependency on limb size. Is the directionality of the relationship from size to Dusp6 expression or is it changes in Dusp6 expression that partially determine limb size?

Our results indicate that there is significant correlation between the limb morphology and the expression of *Dusp6*. However, we cannot conclude that the altered *Dusp6* expression causes the limb defects of Apert syndrome. The limb dysmorphology is likely the result of altered *Fgf* signaling that affects many downstream targets, including *Dusp6*, each with different functions and spatio-temporal patterns. With this method we can identify when phenotype and genetic alterations first appear and whether they are associated or independent from each other. However, we cannot quantify which percentage of the dysmorphology is due to an altered gene expression. We have added a sentence in the Discussion to more clearly explain this issue.

Discussion section. The authors indicated that the approach could be similarly applied to other developmental defects in other organ systems. A discussion of which tissues or organs are particular amenable to this method or may be particularly problematic would be an interesting addition.

The approach is indeed powerful and adaptable to other organs and model systems. The main limitation is the availability of gene probes to perform WMISH and the intrinsic shape of the organ and the gene expression domain to perform GM. Geometric Morphometrics is based on the location of at least a few points that can be reliably located in all the samples. Thus, GM is difficult to apply in structures such as the developing heart, which in its first stages is just a twisting tube. GM is particularly amenable to more complex 3D shapes, such as the facial prominences, the brain and the limbs. In addition, it could be applied in different animals’ models, such as zebra fish, chicken and*Xenopus.*

We hope other investigators will apply our approach and discovering new processes and mechanisms that cannot simply by detected qualitatively by eye. Whole mount in situ hybridization (WMISH) is a technique that is already in use in most molecular labs around the world. Many groups may already have data on gene expression patterns ready to be quantified. Geometric morphometrics (GM) is also a discipline within everyone’s reach, with a long history, plenty of available resources, user-friendly free software and a big community of users (www.morphometrics.org). This discussion can be found in the revised version of the manuscript in the Discussion section,